# Slate: extending Firedrake's domain-specific abstraction to hybridized solvers for geoscience and beyond

Thomas H. Gibson[1], Lawrence Mitchell[2], David A. Ham[1], and Colin J. Cotter[1]

[1]Department of Mathematics, Imperial College London, London, SW7 2AZ, UK
[2]Department of Computer Science, Durham University, Durham, DH1 3LE, UK

**Correspondence:** Thomas H. Gibson (t.gibson15@imperial.ac.uk)

**Abstract.** Within the finite element community, discontinuous Galerkin (DG) and mixed finite element methods have become increasingly popular in simulating geophysical flows. However, robust and efficient solvers for the resulting saddle-point and elliptic systems arising from these discretizations continue to be an on-going challenge. One possible approach for addressing this issue is to employ a method known as hybridization, where the discrete equations are transformed such that classic static condensation and local post-processing methods can be employed. However, it is challenging to implement hybridization as performant parallel code within complex models, whilst maintaining separation of concerns between applications scientists and software experts. In this paper, we introduce a domain-specific abstraction within the Firedrake finite element library that permits the rapid execution of these hybridization techniques within a code-generating framework. The resulting framework composes naturally with Firedrake's solver environment, allowing for the implementation of hybridization and static condensation as runtime-configurable preconditioners via the Python interface to PETSc, petsc4py. We provide examples derived from second order elliptic problems and geophysical fluid dynamics. In addition, we demonstrate that hybridization shows great promise for improving the performance of solvers for mixed finite element discretizations of equations related to large-scale geophysical flows.

## 1  Introduction

The development of simulation software is an increasingly important aspect of modern scientific computing, in the geosciences in particular. Such software requires a vast range of knowledge spanning several disciplines, ranging from applications expertise to mathematical analysis to high-performance computing and low-level code optimization. Software projects developing automatic code generation systems have become quite popular in recent years, as such systems help create a separation of concerns which focuses on a particular complexity independent from the rest. This allows for agile collaboration between computer scientists with hardware and software expertise, computational scientists with numerical algorithm expertise, and domain scientists such as meteorologists, oceanographers and climate scientists. Examples of such projects in the domain of finite element methods include FreeFEM++ (Hecht, 2012), Sundance (Long et al., 2010), the FEniCS Project (Logg et al., 2012a), Feel++ (Prud'Homme et al., 2012), and Firedrake (Rathgeber et al., 2016).

The finite element method (FEM) is a mathematically robust framework for computing numerical solutions of partial differential equations (PDEs) that has become increasingly popular in fluids and solids models across the geosciences, with a formulation that is highly amenable to code-generation techniques. A description of the weak formulation of the PDEs, together with appropriate discrete function spaces, is enough to characterize the finite element problem. Both the FEniCS and Firedrake projects employ the *Unified Form Language* (UFL) (Alnæs et al., 2014) to specify the finite element integral forms and discrete spaces necessary to properly define the finite element problem. UFL is a highly expressive domain-specific language (DSL) embedded in Python, which provides the necessary abstractions for code generation systems.

There are classes of finite element discretizations resulting in discrete systems that can be solved more efficiently by directly manipulating local tensors. For example, the static condensation technique for the reduction of global finite element systems (Guyan, 1965; Irons, 1965) produces smaller globally-coupled linear systems by eliminating interior unknowns to arrive at an equation for the degrees of freedom defined on cell-interfaces only. This procedure is analogous to the point-wise elimination of variables used in staggered finite difference codes, such as the ENDGame dynamical core (Melvin et al., 2010; Wood et al., 2014) of the UK Meteorological Office (Met Office), but requires the local inversion of finite element systems. For finite element discretizations of coupled PDEs, the hybridization technique provides a mechanism for enabling the static condensation of more complex linear systems. First introduced by Fraeijs de Veubeke (1965) and analyzed further by Brezzi and Fortin (1991); Cockburn et al. (2009a); Boffi et al. (2013), the hybridization method introduces Lagrange multipliers enforcing certain continuity constraints. Local static condensation can then be applied to the augmented system to produce a reduced equation for the multipliers. Methods of this type are often accompanied by local post-processing techniques, which exploit the approximation properties of the Lagrange multipliers. This enables the manufacturing of fields exhibiting superconvergent phenomena, or enhanced conservation properties (Arnold and Brezzi, 1985; Brezzi et al., 1985; Bramble and Xu, 1989; Stenberg, 1991; Cockburn et al., 2009b, 2010b). These procedures require invasive manual intervention during the equation assembly process in intricate numerical code.

In this paper, we provide a simple yet effective high-level abstraction for localized dense linear algebra on systems derived from finite element problems. Using embedded DSL technology, we provide a means to enable the rapid development of hybridization and static condensation techniques within an automatic code-generation framework. In other words, the main contribution of this paper is in solving the problem of automatically translating from the mathematics of static condensation and hybridization to compiled code. This automated translation facilitates the separation of concerns between applications scientists and computational/computer scientists, and facilitates the automated optimization of compiled code. This framework provides an environment for the development and testing of numerics relevant to the Gung-Ho Project, an initiative by the UK Met Office in designing the next-generation atmospheric dynamical core using mixed finite element methods (Melvin et al., 2019). Our work is implemented in the Firedrake finite element library and the PETSc solver library (Balay et al., 1997, 2019), accessed via the Python interface petsc4py (Dalcin et al., 2011).

The rest of the paper is organized as follows. We introduce common notation used throughout the paper in Section 1.1. The embedded DSL, called "Slate", is introduced in Section 2, which allows concise expression of localized linear algebra operations on finite element tensors. We provide some contextual examples for static condensation and hybridization in Section

3, including a discussion on post-processing. We then outline in Section 4 how, by interpreting static condensation techniques as a preconditioner, we can go further, and automate many of the symbolic manipulations necessary for hybridization and static condensation. We first demonstrate our implementation on a manufactured problem derived from a second-order elliptic equation, starting in Section 5. The first example compares a hybridizable discontinuous Galerkin (HDG) method with an optimized continuous Galerkin method. Section 5.2 illustrates the composability and relative performance of hybridization for compatible mixed methods applied to a semi-implicit discretization of the nonlinear rotating shallow water equations. Our final example in Section 5.3 demonstrates time-step robustness of a hybridizable solver for a compatible finite element discretization of a rotating linear Boussinesq model. Conclusions follow in Section 6.

## 1.1 Notation

We begin by establishing notation used throughout this paper. Let $\mathcal{T}_h$ denote a tessellation of $\Omega \subset \mathbb{R}^n$, the computational domain, consisting of polygonal elements $K$ associated with a mesh size parameter $h$, and $\partial \mathcal{T}_h = \{e \in \partial K : K \in \mathcal{T}_h\}$ the set of facets of $\mathcal{T}_h$. The set of facets *interior* to the domain $\Omega$ is denoted by $\mathcal{E}_h^\circ := \partial \mathcal{T}_h \setminus \partial \Omega$. Similarly, we denote the set of *exterior* facets as $\mathcal{E}_h^\partial := \partial \mathcal{T}_h \cap \partial \Omega$. For brevity, we denote the finite element integral forms over $\mathcal{T}_h$ and any facet set $\Gamma \subset \partial \mathcal{T}_h$ by

$$(u,v)_K = \int_K u \cdot v \, \mathrm{d}x, \qquad\qquad \langle u,v \rangle_e = \int_e u \cdot v \, \mathrm{d}s, \tag{1}$$

$$(u,v)_{\mathcal{T}_h} = \sum_{K \in \mathcal{T}_h} (u,v)_K, \qquad\qquad \langle u,v \rangle_\Gamma = \sum_{e \in \Gamma} \langle u,v \rangle_e, \tag{2}$$

where $\mathrm{d}x$ and $\mathrm{d}s$ denote appropriate integration measures. The operation $\cdot$ should be interpreted as standard multiplication for scalar functions or a dot product for vector functions.

For any double-valued vector field $\boldsymbol{w}$ on a facet $e \in \partial \mathcal{T}_h$, we define the jump of its normal component across $e$ by

$$[[\boldsymbol{w}]]_e = \begin{cases} \boldsymbol{w}|_{e^+} \cdot \boldsymbol{n}_{e^+} + \boldsymbol{w}|_{e^-} \cdot \boldsymbol{n}_{e^-}, & e \in \mathcal{E}_h^\circ \\ \boldsymbol{w}|_e \cdot \boldsymbol{n}_e, & e \in \mathcal{E}_h^\partial \end{cases} \tag{3}$$

where $+$ and $-$ denote arbitrarily but globally defined sides of the facet. Here, $\boldsymbol{n}_{e^+}$ and $\boldsymbol{n}_{e^-}$ are the unit normal vectors with respect to the positive and negative sides of the facet $e$. Whenever the facet domain is clear by the context, we omit the subscripts for brevity and simply write $[[\cdot]]$.

## 2 A system for localized algebra on finite element tensors

We present an expressive language for dense linear algebra on the elemental matrix systems arising from finite element problems. The language, which we call *Slate*, provides typical mathematical operations performed on matrices and vectors, hence the input syntax is comparable to high-level linear algebra software such as MATLAB. The Slate language provides basic abstract building blocks which can be used by a specialized compiler for linear algebra to generate low-level code implementations.

Slate is heavily influenced by the Unified Form Language (UFL) (Alnæs et al., 2014; Logg et al., 2012a), a DSL embedded in Python which provides symbolic representations of finite element forms. The expressions can be compiled by a *form compiler*, which translates UFL into low level code for the local assembly of a form over the cells and facets of a mesh. In a similar manner, Slate expressions are compiled to low level code that performs the requested linear algebra element-wise on a mesh.

## 2.1 An overview of Slate

To clarify conventions and the scope of Slate, we start by establishing our notation for a general finite element form following the convention of Alnæs et al. (2014). We define a real-valued *multi-linear form* as an operator which maps a list of *arguments* $\boldsymbol{v} = (v_0, \cdots, v_{\alpha-1}) \in V_0 \times \cdots \times V_{\alpha-1}$ into $\mathbb{R}$:

$$a : V_0 \times \cdots \times V_{\alpha-1} \to \mathbb{R}, \quad a \mapsto a(v_0, \cdots, v_{\alpha-1}) = a(\boldsymbol{v}), \tag{4}$$

where $a$ is linear in each argument $v_k$. The *arity* of a form is $\alpha$, an integer denoting the total number of form arguments. In traditional finite element nomenclature (for $\alpha \leq 2$), $V_0$ is referred to as the space of *test functions* and $V_1$ as the space of *trial functions*. Each $V_k$ are referred to as *argument spaces*. Forms with arity $\alpha = 0, 1$ or $2$ are best interpreted as the more familiar mathematical objects: scalars (0-forms), linear forms or functionals (1-forms), and bilinear forms (2-forms) respectively.

If a given form $a$ is parameterized by one or more *coefficients*, say $\boldsymbol{c} = (c_0, \cdots, c_q) \in C_0 \times \cdots \times C_q$ where $\{C_k\}_{k=0}^q$ are *coefficient spaces*, then we write:

$$a : C_0 \times \cdots \times C_q \times V_0 \times \cdots \times V_{\alpha-1} \to \mathbb{R}, \quad a \mapsto a(c_0, \cdots, c_q; v_0, \cdots, v_{\alpha-1}) = a(\boldsymbol{c}; \boldsymbol{v}). \tag{5}$$

From here on, we shall work exclusively with forms that are linear in $\boldsymbol{v}$ and possibly nonlinear in the coefficients $\boldsymbol{c}$. This is reasonable since nonlinear methods based on Newton iterations produce linear problems via Gâteaux differentiation of a nonlinear form corresponding to a PDE (also known as the *form Jacobian*). We refer the interested reader to Alnæs et al. (2014, Section 2.1.2) for more details. For clarity, we present examples of multi-linear forms of arity $\alpha = 0, 1$ and $2$ that frequently appear in finite element discretizations:

$$a(\kappa; v, u) := (\nabla v, \kappa \nabla u)_{\mathcal{T}_h} \equiv \sum_{K \in \mathcal{T}_h} \int_K \nabla v \cdot (\kappa \nabla u) \, \mathrm{d}x, \qquad \kappa \in C_0, \quad u \in V_1, \quad v \in V_0, \quad \alpha = 2, \quad q = 1, \tag{6}$$

$$a(f; v) := (v, f)_{\mathcal{T}_h} \equiv \sum_{K \in \mathcal{T}_h} \int_K v f \, \mathrm{d}x, \qquad f \in C_0, \quad v \in V_0, \quad \alpha = 1, \quad q = 1, \tag{7}$$

$$a(f, g;) := (f - g, f - g)_{\mathcal{T}_h} \equiv \sum_{K \in \mathcal{T}_h} \int_K |f - g|^2 \, \mathrm{d}x, \qquad g \in C_1, \quad f \in C_0, \quad \alpha = 0, \quad q = 2, \tag{8}$$

$$a(\gamma, \boldsymbol{\sigma}) := \langle \gamma, [[\boldsymbol{\sigma}]] \rangle_{\partial \mathcal{T}_h} \equiv \sum_{e \in \mathcal{E}_h^\circ} \int_e \gamma \, [[\boldsymbol{\sigma}]] \, \mathrm{d}s + \sum_{e \in \mathcal{E}_h^\partial} \int_e \gamma \boldsymbol{\sigma} \cdot \boldsymbol{n} \, \mathrm{d}s, \qquad \boldsymbol{\sigma} \in V_1, \quad \gamma \in V_0, \quad \alpha = 2, \quad q = 0. \tag{9}$$

In general, a finite element form will consist of integrals over various geometric domains: integration over cells $\mathcal{T}_h$, interior facets $\mathcal{E}_h^\circ$, and exterior facets $\mathcal{E}_h^\partial$. Therefore, we express a general multi-linear form in terms of integrals over each set of

geometric entities:

$$a(\boldsymbol{c};\boldsymbol{v}) = \sum_{K \in \mathcal{T}_h} \int_K \mathcal{I}_K^{\mathcal{T}}(\boldsymbol{c};\boldsymbol{v})\,\mathrm{d}x + \sum_{e \in \mathcal{E}_h^\circ} \int_e \mathcal{I}_e^{\mathcal{E},\circ}(\boldsymbol{c};\boldsymbol{v})\,\mathrm{d}s + \sum_{e \in \mathcal{E}_h^\partial} \int_e \mathcal{I}_e^{\mathcal{E},\partial}(\boldsymbol{c};\boldsymbol{v})\,\mathrm{d}s, \tag{10}$$

where $\mathcal{I}_K^{\mathcal{T}}$ denotes a cell integrand on $K \in \mathcal{T}_h$, $\mathcal{I}_e^{\mathcal{E},\circ}$ is an integrand on the interior facet $e \in \mathcal{E}_h^\circ$, and $\mathcal{I}_e^{\mathcal{E},\partial}$ is an integrand defined on the exterior facet $e \in \mathcal{E}_h^\partial$. The form $a(\boldsymbol{c};\boldsymbol{v})$ describes a finite element form *globally* over the entire problem domain.

Here, we will consider the case where the interior facet integrands $\mathcal{I}_e^{\mathcal{E},\circ}(\boldsymbol{c};\boldsymbol{v})$ can be decomposed into two independent parts on each interior facet $e$: one for the positive restriction $(+)$ and the negative restriction $(-)$. That is, for each $e \in \mathcal{E}_h^\circ$, we may write: $\mathcal{I}_e^{\mathcal{E},\circ}(\boldsymbol{c};\boldsymbol{v}) = \mathcal{I}_{e+}^{\mathcal{E},\circ}(\boldsymbol{c};\boldsymbol{v}) + \mathcal{I}_{e-}^{\mathcal{E},\circ}(\boldsymbol{c};\boldsymbol{v})$. This allows us to express the integral over an interior facet $e$ connecting two adjacent elements, say $K^+$ and $K^-$, as the sum of integrals:

$$\int_{e \subset \partial K^+ \cup \partial K^-} \mathcal{I}_e^{\mathcal{E},\circ}(\boldsymbol{c};\boldsymbol{v})\,\mathrm{d}s = \int_{e \subset \partial K^+} \mathcal{I}_{e+}^{\mathcal{E},\circ}(\boldsymbol{c};\boldsymbol{v})\,\mathrm{d}s + \int_{e \subset \partial K^-} \mathcal{I}_{e-}^{\mathcal{E},\circ}(\boldsymbol{c};\boldsymbol{v})\,\mathrm{d}s. \tag{11}$$

The local contribution of (10) in each cell $K$, along with its associated facets $e \subset \partial K$, is then

$$a_K(\boldsymbol{c};\boldsymbol{v}) = \int_K \mathcal{I}_K^{\mathcal{T}}(\boldsymbol{c};\boldsymbol{v})\,\mathrm{d}x + \sum_{e \subset \partial K \setminus \partial \Omega} \int_e \mathcal{I}_e^{\mathcal{E},\circ}(\boldsymbol{c};\boldsymbol{v})\,\mathrm{d}s + \sum_{e \subset \partial K \cap \partial \Omega} \int_e \mathcal{I}_e^{\mathcal{E},\partial}(\boldsymbol{c};\boldsymbol{v})\,\mathrm{d}s. \tag{12}$$

We call (12) the *cell-local* contribution of $a(\boldsymbol{c};\boldsymbol{v})$, with

$$a(\boldsymbol{c};\boldsymbol{v}) = \sum_{K \in \mathcal{T}_h} a_K(\boldsymbol{c};\boldsymbol{v}). \tag{13}$$

     To make matters concrete, let us suppose $a(\boldsymbol{c};\boldsymbol{v})$ is a bilinear form with arguments $\boldsymbol{v} = (v_0, v_1) \in V_0 \times V_1$. Now let $\{\Phi_i\}_{i=1}^N$

and $\{\Psi_i\}_{i=1}^M$ denote bases for $V_0$ and $V_1$ respectively. Then the global $N \times M$ matrix $\boldsymbol{A}$ corresponding to $a(\boldsymbol{c};v_0,v_1)$ has its entries defined via

$$\boldsymbol{A}_{ij} = a\left(\boldsymbol{c};\Phi_i,\Psi_j\right) = \sum_{K \in \mathcal{T}_h} \boldsymbol{A}_{K,ij}, \quad \boldsymbol{A}_{K,ij} = a_K\left(\boldsymbol{c};\Phi_i,\Psi_j\right). \tag{14}$$

By construction, $\boldsymbol{A}_{K,ij} \neq \boldsymbol{0}$ if and only if $\Phi_i$ and $\Psi_j$ take non-zero values in $K$. Now we introduce the *cell-node map* $i = e(K,\hat{i})$ as the mapping from the local node number $\hat{i}$ in $K$ to the global node number $i$. Suppose there are $n$ and $m$ nodes

defining the degrees of freedom for $V_0$ and $V_1$, respectively, in $K$. Then all non-zero entries of $\boldsymbol{A}_{K,ij}$ arise from integrals involving basis functions with local indices corresponding to the global indices $i, j$:

$$\boldsymbol{A}_{\hat{i}\hat{j}}^K := a_K\left(\boldsymbol{c};\Phi_{e(K,\hat{i})},\Psi_{e(K,\hat{j})}\right), \quad \hat{i} \in \{1,\cdots,n\}, \quad \hat{j} \in \{1,\cdots,m\}. \tag{15}$$

These local contributions are collected in the $n \times m$ dense matrix $\boldsymbol{A}^K$, which we call the *element tensor*. The global matrix $\boldsymbol{A}$ is assembled from the collection of element tensors: $\boldsymbol{A} \leftarrow \{\boldsymbol{A}^K\}_{K \in \mathcal{T}_h}$. For details on the general evaluation of finite element

basis functions and multi-linear forms, we refer the reader to Kirby (2004); Kirby and Logg (2006); Logg et al. (2012b); Homolya et al. (2018). Further details on the global assembly of finite element operators, with a particular focus on code-generation, are summarized in the work of Logg and Wells (2010); Markall et al. (2013).

In standard finite element software packages, the element tensor is mapped entry-wise into a global sparse array using the cell-node map $e(K, \cdot)$. Within Firedrake, this operation is handled by PyOP2 (Rathgeber et al., 2012) and serves as the main user-facing abstraction for global finite element assembly. For many applications, one may want to produce a new global operator by algebraically manipulating different element tensors. This is relatively invasive in numerical code, as it requires bypassing direct operator assembly to produce the new tensor. This is precisely the scope of Slate.

Like UFL, Slate relies on the grammar of the host-language: Python. The entire Slate language is implemented as a Python module which defines its types (classes) and operations on said types. Together, this forms a high-level language for expressing dense linear algebra on element tensors. The Slate language consists of two primary abstractions for linear algebra:

1. terminal element tensors corresponding to multi-linear integral forms (matrices, vectors, and scalars), or assembled data (for example, coefficient vectors of a finite element function); and

2. expressions consisting of algebraic operations on terminal tensors.

The composition of binary and unary operations on terminal tensors produces a *Slate expression*. Such expressions can be composed with other Slate objects in arbitrary ways, resulting in concise representations of complex algebraic operations on locally assembled arrays. We summarize all currently supported Slate abstractions here.

### 2.1.1 Terminal tensors

In Slate, one associates a tensor with data on a cell either by using a multi-linear form, or assembled coefficient data:

- `Tensor`$(a(\boldsymbol{c}; \boldsymbol{v}))$
  associates a form, expressed in UFL, with its local element tensor:

$$\boldsymbol{A}^K \leftarrow a_K(\boldsymbol{c}; \boldsymbol{v}), \text{ for all } K \in \mathcal{T}_h. \tag{16}$$

  The form arity $\alpha$ of $a_K(\boldsymbol{c}; \boldsymbol{v})$ determines the *rank* of the corresponding `Tensor`, i.e. scalars, vectors, and matrices are produced from scalars, linear forms, and bilinear forms respectively.[1] The *shape* of the element tensor is determined by both the number of arguments, and total number of degrees of freedom local to the cell.

- `AssembledVector`$(f)$
  where $f$ is some finite element function. The function $f \in V$ is expressed in terms of the finite element basis of $V$: $f(x) = \sum_{i=1}^{N} f_i \Phi_i(x)$. The result is the local coefficient vector of $f$ on $K$:

$$\boldsymbol{F}^K \leftarrow \left\{ f_{e(K, \hat{\imath})} \right\}_{\hat{\imath}=1}^{n}, \tag{17}$$

  where $e(K, \hat{\imath})$ is the local node numbering and $n$ is the number of nodes local to the cell $K$.

---

[1] Similarly to UFL, Slate is capable of abstractly representing arbitrary rank tensors. However, only rank $\leq 2$ tensors are typically used in most finite element applications and therefore we currently only generate code for those ranks.

### 2.1.2 Symbolic linear algebra

Slate supports typical binary and unary operations in linear algebra, with a high-level syntax close to mathematics. At the time of this paper, these include:

- `A + B`, the addition of two equal shaped tensors: $\boldsymbol{A}^K + \boldsymbol{B}^K$.

- `A * B`, a contraction over the last index of `A` and the first index of `B`. This is the usual multiplicative operation on matrices, vectors, and scalars: $\boldsymbol{A}^K \boldsymbol{B}^K$.

- `-A`, the additive inverse (negation) of a tensor: $-\boldsymbol{A}^K$.

- `A.T`, the transpose of a tensor: $\left(\boldsymbol{A}^K\right)^T$.

- `A.inv`, the inverse of a square tensor: $\left(\boldsymbol{A}^K\right)^{-1}$.

- `A.solve(B, decomposition="...")`, the result, $\boldsymbol{X}^K$, of solving a local linear system $\boldsymbol{A}^K \boldsymbol{X}^K = \boldsymbol{B}^K$, optionally specifying a factorization strategy.

- `A.blocks[indices]`, where `A` is a tensor from a mixed finite element space. This allows for the extraction of subblocks, which are indexed by field (slices are allowed). For example, if a matrix $A$ corresponds to the bilinear form $a : V \times W \to \mathbb{R}$, where $V = V_0 \times \cdots \times V_n$ and $W = W_0 \times \cdots \times W_m$ are product spaces consisting of finite element spaces $\{V_i\}_{i=0}^n$, $\{W_i\}_{i=0}^m$, then the element tensors have the form:

$$\boldsymbol{A}^K = \begin{bmatrix} \boldsymbol{A}_{00}^K & \boldsymbol{A}_{01}^K & \cdots & \boldsymbol{A}_{0m}^K \\ \boldsymbol{A}_{10}^K & \boldsymbol{A}_{11}^K & \cdots & \boldsymbol{A}_{1m}^K \\ \vdots & \vdots & \ddots & \vdots \\ \boldsymbol{A}_{n0}^K & \boldsymbol{A}_{n1}^K & \cdots & \boldsymbol{A}_{nm}^K \end{bmatrix}. \tag{18}$$

The associated submatrix of (18) with indices $\boldsymbol{i} = (\boldsymbol{p}, \boldsymbol{q})$, $\boldsymbol{p} = \{p_1, \cdots, p_r\}$, $\boldsymbol{q} = \{q_1, \cdots, q_c\}$, is

$$\boldsymbol{A}_{\boldsymbol{pq}}^K = \begin{bmatrix} \boldsymbol{A}_{p_1 q_1}^K & \cdots & \boldsymbol{A}_{p_1 q_c}^K \\ \vdots & \ddots & \vdots \\ \boldsymbol{A}_{p_r q_1}^K & \cdots & \boldsymbol{A}_{p_r q_c}^K \end{bmatrix} = \boldsymbol{A}^K.\texttt{blocks}[\boldsymbol{p}, \boldsymbol{q}], \tag{19}$$

where $\boldsymbol{p} \subseteq \{0, \cdots, n\}$, $\boldsymbol{q} \subseteq \{0, \cdots, m\}$.

Each `Tensor` object knows all the information about the underlying UFL form that defines it, such as form arguments, coefficients, and the underlying finite element space(s) it operates on. This information is propagated through as unary or binary transformations are applied. The unary and binary operations shown here provide the necessary algebraic framework for a large class of problems, some of which we present in this paper.

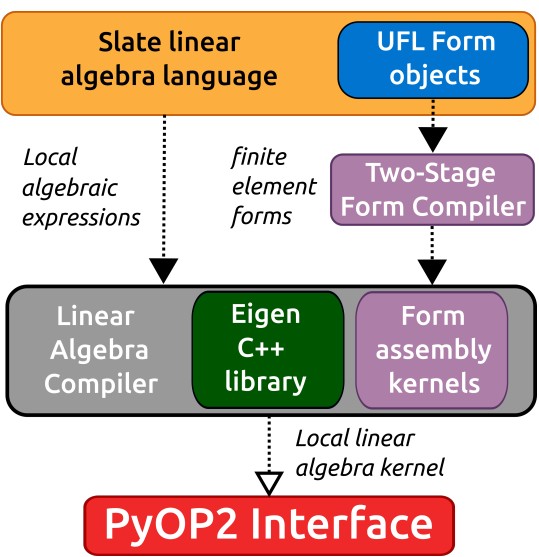

**Figure 1.** The Slate language wraps UFL objects describing the finite element system. The resulting Slate expressions are passed to a specialized linear algebra compiler, which produces a single "macro" kernel assembling the local contributions and executes the dense linear algebra represented in Slate. The kernels are passed to the Firedrake's PyOP2 interface, which wraps the Slate kernel in a mesh-iteration kernel. Parallel scheduling, code generation, and compilation occurs after the PyOP2 layer.

In Firedrake, Slate expressions are transformed into low-level code by a *linear algebra compiler*. The compiler interprets Slate expressions as a *syntax tree*, where the tree is visited to identify what local arrays need to be assembled and the sequence of array operations. At the time of this work, our compiler generates C++ code, using the templated library Eigen (Guennebaud et al., 2015) for dense linear algebra. The translation from Slate to C++ is fairly straightforward, as all operations supported by
Slate have a representation in Eigen.

The compiler pass will generate a single "macro" kernel, which performs the dense linear algebra operations represented in Slate. The resulting code will also include (often multiple) function calls to local assembly kernels generated by TSFC (Homolya et al., 2018) to assemble all necessary sub-blocks of an element tensor. All code generated by the linear algebra compiler conforms to the application programming interface (API) of the PyOP2 framework, as detailed by Rathgeber et al.
(2012, Section 3). Figure 1 provides an illustration of the complete tool-chain.

Most optimization of the resulting dense linear algebra code is handled directly by Eigen. In the case of unary and binary operations such as `A.inv` and `A.solve(B)`, stable default behaviors are applied by the linear algebra compiler. For example, `A.solve(B)` without a specified factorization strategy will default to using an in-place LU factorization with partial pivoting. For local matrices smaller than $5 \times 5$, the inverse is translated directly into Eigen's `A.inverse()` which employs stable
analytic formulas. For larger matrices, the linear algebra replaces `A.inv` with an LU factorization.[2] Currently, we only support

---

[2]For more details on solving linear equations in Eigen, see: https://eigen.tuxfamily.org/dox/group__TutorialLinearAlgebra.html

direct matrix factorizations for solving local linear systems. However, it would not be difficult to extend Slate to support more general solution techniques like iterative methods.

## 3 Examples

We now present examples and discuss solution methods which require element-wise manipulations of finite element systems and their specification in Slate. We stress here that Slate is not limited to these model problems; rather these examples are chosen for clarity and to demonstrate key features of the Slate language. For our discussion, we use a model elliptic equation defined in a computational domain $\Omega$. Consider the second-order PDE with both Dirichlet and Neumann boundary conditions:

$$-\nabla \cdot (\kappa \nabla p) + cp = f, \quad \text{in } \Omega, \tag{20}$$

$$p = p_0, \quad \text{on } \partial\Omega_D, \tag{21}$$

$$-\kappa \nabla p \cdot \boldsymbol{n} = g, \quad \text{on } \partial\Omega_N, \tag{22}$$

where $\partial\Omega_D \cup \partial\Omega_N = \partial\Omega$ and $\kappa, c : \Omega \to \mathbb{R}^+$ are positive-valued coefficients. To obtain a mixed formulation of (20)–(22), we introduce the auxiliary velocity variable $\boldsymbol{u} = -\kappa \nabla p$. We then obtain the first-order system of PDEs:

$$\mu \boldsymbol{u} + \nabla p = 0, \quad \text{in } \Omega, \tag{23}$$

$$\nabla \cdot \boldsymbol{u} + cp = f, \quad \text{in } \Omega, \tag{24}$$

$$p = p_0, \quad \text{on } \partial\Omega_D, \tag{25}$$

$$\boldsymbol{u} \cdot \boldsymbol{n} = g, \quad \text{on } \partial\Omega_N, \tag{26}$$

where $\mu = \kappa^{-1}$.

### 3.1 Hybridization of mixed methods

To motivate our discussion in this section, we start by recalling the mixed method for (23)–(26). Methods of this type seek approximations $(\boldsymbol{u}_h, p_h)$ in finite-dimensional subspaces $U_h \times V_h \subset H(\text{div}; \Omega) \times L^2(\Omega)$, defined by:

$$U_h = \{\boldsymbol{w} \in H(\text{div}; \Omega) : \boldsymbol{w}|_K \in U(K), \forall K \in \mathcal{T}_h, \boldsymbol{w} \cdot \boldsymbol{n} = g \text{ on } \partial\Omega_N\}, \tag{27}$$

$$V_h = \{\phi \in L^2(\Omega) : \phi|_K \in V(K), \forall K \in \mathcal{T}_h\}. \tag{28}$$

The space $U_h$ consists of $H(\text{div})$-conforming piecewise vector polynomials, where choices of $U(K)$ typically include the Raviart-Thomas (RT), Brezzi-Douglas-Marini (BDM), or Brezzi-Douglas-Fortin-Marini (BDFM) elements (Raviart and Thomas, 1977; Nédélec, 1980; Brezzi et al., 1985, 1987). The space $V_h$ is the Lagrange family of discontinuous polynomials. These spaces are of particular interest when simulating geophysical flows, since choosing the right pairing results in stable discretizations with desirable conservation properties and avoids spurious computational modes. We refer the reader to Cotter

and Shipton (2012); Cotter and Thuburn (2014); Natale et al. (2016); Shipton et al. (2018) for a discussion of mixed methods relevant for geophysical fluid dynamics. Two examples of such discretizations are presented in Section 5.2.

The mixed formulation of (23)–(26) is arrived at by multiplying (23)–(24) by test functions and integrating by parts. The resulting finite element problem reads as follows: find $(\boldsymbol{u}_h, p_h) \in U_h \times V_h$ satisfying

$$
(\boldsymbol{w}, \mu\boldsymbol{u}_h)_{\mathcal{T}_h} - (\nabla \cdot \boldsymbol{w}, p_h)_{\mathcal{T}_h} = -\langle \boldsymbol{w} \cdot \boldsymbol{n}, p_0 \rangle_{\partial\Omega_D}, \quad \forall \boldsymbol{w} \in U_{h,0}, \tag{29}
$$

$$
(\phi, \nabla \cdot \boldsymbol{u}_h)_{\mathcal{T}_h} + (\phi, cp_h)_{\mathcal{T}_h} = (\phi, f)_{\mathcal{T}_h}, \quad \forall \phi \in V_h, \tag{30}
$$

where $U_{h,0}$ is the subspace of $U_h$ with functions whose normal components vanish on $\partial\Omega_N$. The discrete system is obtained by first expanding the solutions in terms of the finite element bases:

$$
\boldsymbol{u}_h = \sum_{i=1}^{N_{\boldsymbol{u}}} U_i \boldsymbol{\Psi}_i, \quad p_h = \sum_{i=1}^{N_p} P_i \xi_i, \tag{31}
$$

where $\{\boldsymbol{\Psi}_i\}_{i=1}^{N_{\boldsymbol{u}}}$ and $\{\xi_i\}_{i=1}^{N_p}$ are bases for $U_h$ and $V_h$ respectively. Here, $U_i$ and $P_i$ are the coefficients to be determined. As per standard Galerkin-based finite element methods, taking $\boldsymbol{w} = \boldsymbol{\Psi}_j$, $j \in \{1, \cdots, N_{\boldsymbol{u}}\}$ and $\phi = \xi_j$, $j \in \{1, \cdots, N_p\}$ in (29)–(30) produces the discrete saddle point system:

$$
\begin{bmatrix} \boldsymbol{A} & -\boldsymbol{B}^T \\ \boldsymbol{B} & \boldsymbol{D} \end{bmatrix} \begin{Bmatrix} \boldsymbol{U} \\ \boldsymbol{P} \end{Bmatrix} = \begin{Bmatrix} \boldsymbol{F}_0 \\ \boldsymbol{F}_1 \end{Bmatrix}. \tag{32}
$$

where $\boldsymbol{U} = \{U_i\}_{i=1}^{N_{\boldsymbol{u}}}$, $\boldsymbol{P} = \{P_i\}_{i=1}^{N_p}$ are the coefficient vectors, and

$$
\boldsymbol{A}_{ij} = (\boldsymbol{\Psi}_i, \mu\boldsymbol{\Psi}_j)_{\mathcal{T}_h}, \tag{33}
$$

$$
\boldsymbol{B}_{ij} = (\xi_i, \nabla \cdot \boldsymbol{\Psi}_j)_{\mathcal{T}_h}, \tag{34}
$$

$$
\boldsymbol{D}_{ij} = (\xi_i, c\xi_j)_{\mathcal{T}_h}, \tag{35}
$$

$$
\boldsymbol{F}_{0,j} = -\langle \boldsymbol{\Psi}_j \cdot \boldsymbol{n}, p_0 \rangle_{\partial\Omega_D}, \tag{36}
$$

$$
\boldsymbol{F}_{1,j} = (\xi_j, f)_{\mathcal{T}_h}. \tag{37}
$$

Methods to efficiently invert such systems include $H(\mathrm{div})$-multigrid (Arnold et al., 2000) (requiring complex overlapping-Schwarz smoothers), global Schur-complement factorizations (which require an approximation to the inverse of the *dense*[3] elliptic Schur-complement $\boldsymbol{D} + \boldsymbol{B}\boldsymbol{A}^{-1}\boldsymbol{B}^T$), or auxiliary space multigrid (Hiptmair and Xu, 2007). Here, we focus on a solution approach using a hybridized mixed method (Arnold and Brezzi, 1985; Brezzi and Fortin, 1991; Boffi et al., 2013).

The hybridization technique replaces the original system with a discontinuous variant, decoupling the velocity degrees of freedom between cells. This is done by replacing the discrete solution space for $\boldsymbol{u}_h$ with the "broken" space $U_h^d$, defined as:

$$
U_h^d = \{\boldsymbol{w} \in [L^2(\Omega)]^n : \boldsymbol{w}|_K \in U(K), \forall K \in \mathcal{T}_h\}. \tag{38}
$$

---

[3]The Schur-complement, while elliptic, is globally dense due to the fact that $\boldsymbol{A}$ has a dense inverse. This is a result of velocities in $U_h$ having continuous normal components across cell-interfaces.

The vector finite element space $U_h^d$ is a subspace of $[L^2(\Omega)]^n$ consisting of local $H(\mathrm{div})$ functions, but normal components are no longer required to be continuous on $\partial \mathcal{T}_h$. The approximation space for $p_h$ remains unchanged.

Next, Lagrange multipliers are introduced as an auxiliary variable in the space $M_h$, defined only on cell-interfaces:

$$M_h = \{\gamma \in L^2(\partial \mathcal{T}_h) : \gamma|_e \in M(e), \forall e \in \partial \mathcal{T}_h\}, \tag{39}$$

where $M(e)$ denotes a polynomial space defined on each facet. We call $M_h$ the space of approximate traces. Functions in $M_h$ are discontinuous across vertices in two-dimensions, and vertices/edges in three-dimensions.

Deriving the hybridizable mixed system is accomplished through integration by parts over each element $K$. Testing with $\boldsymbol{w} \in U_h^d(K)$ and integrating (23) over the cell $K$ produces:

$$\left(\boldsymbol{w}, \mu \boldsymbol{u}_h^d\right)_K - (\nabla \cdot \boldsymbol{w}, p_h)_K + \langle \boldsymbol{w} \cdot \boldsymbol{n}, \lambda_h \rangle_{\partial K} = -\langle \boldsymbol{w} \cdot \boldsymbol{n}, p_0 \rangle_{\partial K \cap \partial \Omega_D}. \tag{40}$$

The trace function $\lambda_h$ is introduced in the surface integral as an approximation to $p|_{\partial K}$. An additional constraint equation, called the *transmission condition*, is added to close the system. The resulting hybridizable formulation reads: find $(\boldsymbol{u}_h^d, p_h, \lambda_h) \in U_h^d \times V_h \times M_h$ such that

$$\left(\boldsymbol{w}, \mu \boldsymbol{u}_h^d\right)_{\mathcal{T}_h} - (\nabla \cdot \boldsymbol{w}, p_h)_{\mathcal{T}_h} + \langle [[\boldsymbol{w}]], \lambda_h \rangle_{\partial \mathcal{T}_h \setminus \partial \Omega_D} = -\langle \boldsymbol{w} \cdot \boldsymbol{n}, p_0 \rangle_{\partial \Omega_D}, \quad \forall \boldsymbol{w} \in U_h^d, \tag{41}$$

$$\left(\phi, \nabla \cdot \boldsymbol{u}_h^d\right)_{\mathcal{T}_h} + (\phi, c p_h)_{\mathcal{T}_h} = (\phi, f)_{\mathcal{T}_h}, \quad \forall \phi \in V_h, \tag{42}$$

$$\langle \gamma, [[\boldsymbol{u}_h^d]] \rangle_{\partial \mathcal{T}_h \setminus \partial \Omega_D} = \langle \gamma, g \rangle_{\partial \Omega_N}, \quad \forall \gamma \in M_{h,0}, \tag{43}$$

where $M_{h,0}$ denotes the space of traces vanishing on $\partial \Omega_D$. The transmission condition (43) enforces both continuity of $\boldsymbol{u}_h^d \cdot \boldsymbol{n}$ across element boundaries, as well as the boundary condition: $\boldsymbol{u}_h^d \cdot \boldsymbol{n} = g$ on $\partial \Omega_N$. If the space of Lagrange multipliers $M_h$ is chosen appropriately, then the "broken" velocity $\boldsymbol{u}_h^d$, albeit sought a priori in a discontinuous space, will coincide with its $H(\mathrm{div})$-conforming counterpart. Specifically, the formulations in (41)–(42) and (29)–(30) are solving equivalent problems if the normal components of $\boldsymbol{w} \in U_h$ lie in the same polynomial space as the trace functions (Arnold and Brezzi, 1985).

The discrete matrix system arising from (41)–(43) has the general form:

$$\begin{bmatrix} \boldsymbol{A}_{00} & \boldsymbol{A}_{01} & \boldsymbol{A}_{02} \\ \boldsymbol{A}_{10} & \boldsymbol{A}_{11} & \boldsymbol{A}_{12} \\ \boldsymbol{A}_{20} & \boldsymbol{A}_{21} & \boldsymbol{A}_{22} \end{bmatrix} \begin{Bmatrix} \boldsymbol{U}^d \\ \boldsymbol{P} \\ \boldsymbol{\Lambda} \end{Bmatrix} = \begin{Bmatrix} \boldsymbol{F}_0 \\ \boldsymbol{F}_1 \\ \boldsymbol{F}_2 \end{Bmatrix}, \tag{44}$$

where the discrete system is produced by expanding functions in terms of the finite element bases for $U_h^d$, $V_h$, and $M_h$ like before. Upon initial inspection, it may not appear to be advantageous to replace our original formulation with this augmented equation-set; the hybridizable system has substantially more total degrees of freedom. However, (44) has a considerable advantage over (32) in the following ways:

1. Since both $U_h^d$ and $V_h$ are discontinuous spaces, $\boldsymbol{U}^d$ and $\boldsymbol{P}$ are coupled only within the cell. This allows us to simultaneously eliminate both unknowns via *local* static condensation to produce a significantly smaller global (hybridized)

problem for the trace unknowns, $\mathbf{\Lambda}$:

$$S\mathbf{\Lambda} = E, \tag{45}$$

where $S \leftarrow \{S^K\}_{K \in \mathcal{T}_h}$ and $E \leftarrow \{E^K\}_{K \in \mathcal{T}_h}$ are assembled via the local element tensors:

$$S^K = A_{22}^K - \begin{bmatrix} A_{20}^K & A_{21}^K \end{bmatrix} \begin{bmatrix} A_{00}^K & A_{01}^K \\ A_{10}^K & A_{11}^K \end{bmatrix}^{-1} \begin{bmatrix} A_{02}^K \\ A_{12}^K \end{bmatrix}, \tag{46}$$

$$E^K = F_2^K - \begin{bmatrix} A_{20}^K & A_{21}^K \end{bmatrix} \begin{bmatrix} A_{00}^K & A_{01}^K \\ A_{10}^K & A_{11}^K \end{bmatrix}^{-1} \begin{Bmatrix} F_0^K \\ F_1^K \end{Bmatrix}. \tag{47}$$

Note that the inverse of the block matrix in (46) and (47) is *never* evaluated globally; the elimination can be performed locally by performing a sequence of Schur-complement reductions within each cell.

2. The matrix $S$ is sparse, symmetric, positive-definite, and spectrally equivalent to the dense Schur-complement $D + BA^{-1}B^T$ from (32) of the original mixed formulation (Gopalakrishnan, 2003; Cockburn et al., 2009a).

3. Once $\mathbf{\Lambda}$ is computed, both $U^d$ and $P$ can be recovered locally in each element. This can be accomplished in a number ways. One way is to compute $P^K$ by solving:

$$\left( A_{11}^K - A_{10}^K \left( A_{00}^K \right)^{-1} A_{01}^K \right) P^K = F_1^K - A_{10}^K \left( A_{00}^K \right)^{-1} F_0^K - \left( A_{12}^K - A_{10}^K \left( A_{00}^K \right)^{-1} A_{02}^K \right) \mathbf{\Lambda}^K, \tag{48}$$

followed by solving for $\left( U^d \right)^K$:

$$A_{00}^K \left( U^d \right)^K = F_0^K - A_{01}^K P^K - A_{02}^K \mathbf{\Lambda}^K. \tag{49}$$

Similarly, one could rearrange the order in which each variable is reconstructed.

4. If desired, the solutions can be improved further through local post-processing. We highlight two such procedures for $U^d$ and $P$, respectively, in Section 3.3.

Figure 2 displays the corresponding Slate code for assembling the trace system, solving (45), and recovering the eliminated unknowns. For a complete reference on how to formulate the hybridized mixed system (41)–(43) in UFL, we refer the reader to Alnæs et al. (2014). Complete Firedrake code using Slate to solve a hybridizable mixed system is also publicly available in Zenodo/Tabula-Rasa (2019, "Code verification"). We remark that, in the case of this hybridizable system, (44) contains zero-valued blocks which can simplify the resulting expressions in (46)–(47) and (48)–(49). This is not true in general and therefore the expanded form using all sub-blocks of (44) is presented for completeness.

```
# Element tensors defining the local 3-by-3 block system
_A = Tensor(a)
_F = Tensor(L)
# Extracting blocks for Slate expression of the reduced system
A = _A.blocks
F = _F.blocks
Sexp = A[2, 2] - A[2, :2] * A[:2, :2].inv * A[:2, 2]    # Slate expression for S^K
Eexp = F[2] - A[2, :2] * A[:2, :2].inv * F[:2]          # Slate expression for E^K
S = assemble(Sexp, bcs=[...])                          # Assemble S
E = assemble(Eexp)                                     # Assemble E
lambda_h = Function(M)                                 # Function to store the result: Λ
# Solve for the Lagrange multipliers: Λ
solve(S, lambda_h, E, solver_parameters={"ksp_type": "preonly", "pc_type": "lu"})
p_h = Function(V)                                      # Function to store the result: P
u_h = Function(U)                                      # Function to store the result: U^d
Lambda = AssembledVector(lambda_h)                    # Local coefficient vector: Λ^K
P = AssembledVector(p_h)                              # Local coefficient vector: P^K
# Intermediate expressions
Sd = A[1, 1] - A[1, 0] * A[0, 0].inv * A[0, 1]
Sl = A[1, 2] - A[1, 0] * A[0, 0].inv * A[0, 2]
# Slate expressions for local recovery
p_sys = Sd.solve(F[1] - A[1, 0] * A[0, 0].inv * F[0] - Sl * Lambda,
decomposition="PartialPivLu")
u_sys = A[0, 0].solve(F[0] - A[0, 1] * P - A[0, 2] * Lambda,
decomposition="PartialPivLu")
assemble(p_sys, p_h)                                   # Solve for P
assemble(u_sys, u_h)                                   # Solve for U^d
```

**Figure 2.** Firedrake code for solving (44) via static condensation and local recovery, given UFL expressions a, L for (41)–(43). Arguments of the mixed space $U_h^d \times V_h \times M_h$ are indexed by 0, 1, and 2 respectively. Lines 8 and 9 are symbolic expressions for (46) and (47) respectively. Any vanishing conditions on the trace variables should be provided as boundary conditions during operator assembly (line 10). Lines 26 and 28 are expressions for (48) and (49) (using LU). Code-generation occurs in lines 10, 11, 30, and 31. A global linear solver for the reduced system is created and used in line 15. Configuring the linear solver is done by providing an appropriate Python dictionary of solver options for the PETSc library.

### 3.2 Hybridization of discontinuous Galerkin methods

The hybridized discontinuous Galerkin (HDG) method is a natural extension of discontinuous Galerkin (DG) discretizations. Here, we consider a specific HDG discretization, namely the LDG-H method (Cockburn et al., 2010b). Other forms of HDG that involve local lifting operators can also be implemented in this software framework by the introduction of additional local (i.e., discontinuous) variables in the definition of the local solver.

Deriving the LDG-H formulation follows exactly from standard DG methods. All prognostic variables are sought in the discontinuous spaces $U_h \times V_h \subset [L^2(\Omega)]^n \times L^2(\Omega)$. Within a cell $K$, integration by parts yields:

$$(\boldsymbol{w}, \mu \boldsymbol{u}_h)_K - (\nabla \cdot \boldsymbol{w}, p_h)_K + \langle \boldsymbol{w} \cdot \boldsymbol{n}, \widehat{p} \rangle_{\partial K} = 0, \quad \forall \boldsymbol{w} \in U(K), \tag{50}$$

$$-(\nabla \phi, \boldsymbol{u}_h)_K + \langle \phi, \widehat{\boldsymbol{u}} \cdot \boldsymbol{n} \rangle_{\partial K} + (\phi, c p_h)_K = (\phi, f)_K, \quad \forall \phi \in V(K), \tag{51}$$

where $U(K)$ and $V(K)$ are vector and scalar polynomial spaces respectively. Now, we define the numerical fluxes $\widehat{p}$ and $\widehat{\boldsymbol{u}}$ to be functions of the trial unknowns and a new independent unknown in the trace space $M_h$:

$$\widehat{\boldsymbol{u}}(\boldsymbol{u}_h, p_h, \lambda_h; \tau) = \boldsymbol{u}_h + \tau (p_h - \widehat{p}) \boldsymbol{n}, \tag{52}$$

$$\widehat{p}(\lambda_h) = \lambda_h, \tag{53}$$

where $\lambda_h \in M_h$ is a function approximating $p$ on $\partial \mathcal{T}_h$ and $\tau$ is a positive stabilization function that may vary on each facet $e \in \partial \mathcal{T}_h$. We further require that $\lambda_h$ satisfies the Dirichlet condition for $p$ on $\partial \Omega_D$ in an $L^2$-projection sense. The full LDG-H formulation reads as follows. Find $(\boldsymbol{u}_h, p_h, \lambda_h) \in U_h \times V_h \times M_h$ such that

$$(\boldsymbol{w}, \mu \boldsymbol{u}_h)_{\mathcal{T}_h} - (\nabla \cdot \boldsymbol{w}, p_h)_{\mathcal{T}_h} + \langle [[\boldsymbol{w}]], \lambda_h \rangle_{\partial \mathcal{T}_h} = 0, \quad \forall \boldsymbol{w} \in U_h, \tag{54}$$

$$-(\nabla \phi, \boldsymbol{u}_h)_{\mathcal{T}_h} + \langle \phi, [[\boldsymbol{u}_h + \tau (p_h - \lambda_h) \boldsymbol{n}]] \rangle_{\partial \mathcal{T}_h} + (\phi, c p_h)_{\mathcal{T}_h} = (\phi, f)_{\mathcal{T}_h}, \quad \forall \phi \in V_h, \tag{55}$$

$$\langle \gamma, [[\boldsymbol{u}_h + \tau (p_h - \lambda_h) \boldsymbol{n}]] \rangle_{\partial \mathcal{T}_h \backslash \partial \Omega_D} = \langle \gamma, g \rangle_{\partial \Omega_N}, \quad \forall \gamma \in M_h, \tag{56}$$

$$\langle \gamma, \lambda_h \rangle_{\partial \Omega_D} = \langle \gamma, p_0 \rangle_{\partial \Omega_D}, \quad \forall \gamma \in M_h, \tag{57}$$

Equation (56) is the transmission condtion, which enforces the continuity of $\widehat{\boldsymbol{u}} \cdot \boldsymbol{n}$ on $\partial \mathcal{T}_h$ and (57) ensures $\lambda_h$ satisfies the Dirichlet condition. This ensures that the numerical flux is single-valued on the facets. Hence, the LDG-H method defines a *conservative* DG method (Cockburn et al., 2010b). Note that the choice of $\tau$ has a significant influence on the expected convergence rates of the computed solutions.

The LDG-H method retains the advantages of standard DG methods while also enabling the assembly of reduced linear systems through static condensation. The matrix system arising from (54)–(57) has the same general form as the hybridized mixed method in (44), except all sub-blocks are now populated with non-zero entries due to the coupling of trace functions with both $p_h$ and $\boldsymbol{u}_h$. However, all previous properties of the discrete matrix system from Section 3.1 still apply. The Slate expressions for the local elimination and reconstruction operations will be identical to those illustrated in Figure 2. For the interested reader, a unified analysis of hybridization methods (both mixed and DG) for second-order elliptic equations is presented in Cockburn et al. (2009a); Cockburn (2016).

### 3.3 Local post-processing

For both mixed (Arnold and Brezzi, 1985; Brezzi et al., 1985; Bramble and Xu, 1989; Stenberg, 1991) and discontinuous Galerkin methods (Cockburn et al., 2010b, 2009b), it is possible to locally post-process solutions to obtain superconvergent approximations (gaining one order of accuracy over the unprocessed solution). These methods can be expressed as local solves

on each element and are straightforward to implement using Slate. In this section, we present two post-processing techniques: one for scalar fields, and another for the vector unknown. The Slate code follows naturally from previous discussions in Sections 3.1 and 3.2, using the standard set of operations on element tensors summarized in Section 2.1.

### 3.3.1 Post-processing of the scalar solution

Our first example is a modified version of the procedure presented by Stenberg (1991) for enhancing the accuracy of the scalar solution. This was also highlighted within the context of hybridizing eigenproblems by Cockburn et al. (2010a). This post-processing technique can be used for both the hybridizable mixed and LDG-H methods. We proceed by posing the finite element systems cell-wise.

Let $\mathcal{P}_k(K)$ denote a polynomial space of degree $\leq k$ on a cell $K \in \mathcal{T}_h$. Then for a given pair of computed solutions $\boldsymbol{u}_h, p_h$
of the hybridized methods, we define the post-processed scalar $p_h^\star \in \mathcal{P}_{k+1}(K)$ as the unique solution of the local problem:

$$\left(\nabla w, \nabla p_h^\star\right)_K = -\left(\nabla w, \kappa^{-1}\boldsymbol{u}_h\right)_K, \quad \forall w \in \mathcal{P}_{k+1}^{\perp,l}(K), \tag{58}$$

$$\left(v, p_h^\star\right)_K = (v, p_h)_K, \quad \forall v \in \mathcal{P}_l(K), \tag{59}$$

where $0 \leq l \leq k$. Here, the space $\mathcal{P}_{k+1}^{\perp,l}(K)$ denotes the $L^2$-orthogonal complement of $\mathcal{P}_l(K)$. This post-processing method directly uses the definition of the flux $\boldsymbol{u}_h$, the approximation of $-\kappa\nabla p$. In practice, the space $\mathcal{P}_{k+1}^{\perp,l}(K)$ may be constructed
using an orthogonal hierarchical basis, and solving (58)–(59) amounts to inverting a symmetric positive definite system in each cell of the mesh.

At the time of this work, Firedrake does not support the construction of such a finite element basis. However, we can introduce Lagrange multipliers to enforce the orthogonality constraint. The resulting local problem then becomes the following mixed system: find $(p_h^\star, \psi) \in \mathcal{P}_{k+1}(K) \times \mathcal{P}_l(K)$ such that

$$20 \quad \left(\nabla w, \nabla p_h^\star\right)_K + (w, \psi)_K = -\left(\nabla w, \kappa^{-1}\boldsymbol{u}_h\right)_K, \quad \forall w \in \mathcal{P}_{k+1}(K), \tag{60}$$

$$\left(\phi, p_h^\star\right)_K = (\phi, p_h)_K, \quad \forall \phi \in \mathcal{P}_l(K), \tag{61}$$

where $0 \leq l \leq k$. The local problems (60)–(61) and (58)–(59) are equivalent, with the Lagrange multiplier $\psi$ enforcing orthogonality of test functions in $\mathcal{P}_{k+1}(K)$ with functions in $\mathcal{P}_l(K)$.

This post-processing method produces a new approximation which superconverges at a rate of $k+2$ for hybridized mixed
methods (Stenberg, 1991; Cockburn et al., 2010a). For the LDG-H method, $k+2$ superconvergence is achieved when $\tau = \mathcal{O}(1)$ and $\tau = \mathcal{O}(h)$, but only $k+1$ convergence is achieved when $\tau = \mathcal{O}(1/h)$ (Cockburn et al., 2009b, 2010b). We demonstrate the increased accuracy in computed solutions in Section 5.1. An abridged example using Firedrake and Slate to solve the local linear systems is provided in Figure 3.

```
# Define spaces for the higher-order pressure approximation and Lagrange multipliers
DGk1 = FunctionSpace(mesh, "DG", degree + 1)
DG0 = FunctionSpace(mesh, "DG", 0)
W = DGk1 * DG0
p, psi = TrialFunctions(W)
w, phi = TestFunctions(W)
# Create local Slate tensors for the post-processing system
K = Tensor((inner(grad(p), grad(w)) + inner(psi, w) + inner(p, phi))*dx)
# Use the computed pressure p_h and flux u_h in the right-hand side
F = Tensor((-inner(u_h, grad(w)) + inner(p_h, phi))*dx)
E = K.inv * F
# Function for the post-processed scalar p_h^*
p_star = Function(DGk1, name="Post-processed scalar")
assemble(E.blocks[0], p_star)        # Assemble only the first field (pressure)
```

**Figure 3.** Example of local post-processing using Firedrake and Slate. Here, we locally solve the mixed system defined in (58)–(59). The corresponding symbolic local tensors are defined in lines 9 and 11. The Slate expression for directly inverting the local system is written in line 12. In line 16, a Slate-generated kernel is produced which solves the resulting linear system in each cell. Since we are not interested in the multiplier, we only return the block corresponding to the new pressure field.

### 3.3.2 Post-processing of the flux

Our second example illustrates a procedure that uses the numerical flux of an HDG discretization for (23)–(26). Within the context of the LDG-H method, we can use the numerical trace in (52) to produce a vector field that is $H(\text{div})$-conforming. The technique we outline here follows that of Cockburn et al. (2009b).

Let $\mathcal{T}_h$ be a mesh consisting of simplices. On each cell $K \in \mathcal{T}_h$, we define a new function $\boldsymbol{u}_h^\star$ to be the unique element of the local Raviart-Thomas space $[\mathcal{P}_k(K)]^n + \boldsymbol{x}\mathcal{P}_k(K)$ satisfying

$$(\boldsymbol{r}, \boldsymbol{u}_h^\star)_K = (\boldsymbol{r}, \boldsymbol{u}_h)_K, \quad \forall \boldsymbol{r} \in [\mathcal{P}_{k-1}(K)]^n, \tag{62}$$

$$\langle \mu, \boldsymbol{u}_h^\star \cdot \boldsymbol{n} \rangle_e = \langle \mu, \widehat{\boldsymbol{u}} \cdot \boldsymbol{n} \rangle_e, \quad \forall \mu \in \mathcal{P}_k(e), \tag{63}$$

for all facets $e$ on $\partial K$, where $\widehat{\boldsymbol{u}}$ is the numerical flux defined in (52). This local problem produces a new velocity $\boldsymbol{u}_h^\star$ with the following properties:

1. $\boldsymbol{u}_h^\star$ converges at the *same* rate as $\boldsymbol{u}_h$ for all choices of $\tau$ producing a solvable system for (54)–(57). However,

2. $\boldsymbol{u}_h^\star \in H(\text{div}; \Omega)$. That is, $[[\boldsymbol{u}_h^\star]]_e = 0, \forall e \in \mathcal{E}_h^\circ$.

3. Additionally, the divergence of $\boldsymbol{u}_h^\star$ convergences at a rate of $k + 1$.

The Firedrake implementation using Slate is similar to the scalar post-processing example (see Figure 3); the cell-wise linear systems (62)–(63) can be expressed in UFL, and therefore the necessary Slate expressions to invert the local systems follows naturally from the set of operations presented in Section 2.1. We use the very sensitive parameter dependency in the post-processing methods to validate our software implementation in Zenodo/Tabula-Rasa (2019, "Code-verification").

## 4  Static condensation as a preconditioner

Slate enables static condensation approaches to be expressed very concisely. Nonetheless, application of a particular approach to different variational problems using Slate still requires a certain amount of code repetition. By formulating each form of static condensation as a preconditioner, code can be written once and then applied to any mathematically suitable problem. Rather than writing the static condensation by hand, in many cases, it is sufficient to just select the appropriate, Slate-based, preconditioner.

For context, it is helpful to frame the problem in the particular context of the solver library: PETSc. Firedrake uses PETSc as its main solver abstraction framework, and can provide *operator-based preconditioners* for solving linear systems as `PC` objects expressed in Python via petsc4py (Dalcin et al., 2011). For a comprehensive overview on solving linear systems using PETSc, we refer the interested reader to Balay et al. (2019, Chapter 4).

Suppose we wish to solve a linear system: $\boldsymbol{A}\boldsymbol{x} = \boldsymbol{b}$. We can think of (left) preconditioning the system in residual form:

$$\boldsymbol{r} = \boldsymbol{r}(\boldsymbol{A}, \boldsymbol{b}) \equiv \boldsymbol{b} - \boldsymbol{A}\boldsymbol{x} = \boldsymbol{0} \tag{64}$$

by an operator $\mathcal{P}$ (which may not necessarily be linear) as a transformation into an equivalent system of the form

$$\mathcal{P}\boldsymbol{r} = \mathcal{P}\boldsymbol{b} - \mathcal{P}\boldsymbol{A}\boldsymbol{x} = \boldsymbol{0}. \tag{65}$$

Given a current iterate $\boldsymbol{x}_i$ the residual at the $i$-th iteration is simply $\boldsymbol{r}_i \equiv \boldsymbol{b} - \boldsymbol{A}\boldsymbol{x}_i$, and $\mathcal{P}$ acts on the residual to produce an approximation to the error $\boldsymbol{\epsilon}_i \equiv \boldsymbol{x} - \boldsymbol{x}_i$. If $\mathcal{P}$ is an application of an exact inverse, the residual is converted into an exact (up to numerical round-off) error.

We will denote the application of a particular Krylov subspace method (`KSP`) for the linear system (64) as $\mathcal{K}_{\boldsymbol{x}}(\boldsymbol{r}(\boldsymbol{A}, \boldsymbol{b}))$. Upon preconditioning the system via $\mathcal{P}$ as in (65), we write

$$\mathcal{K}_{\boldsymbol{x}}(\mathcal{P}\boldsymbol{r}(\boldsymbol{A}, \boldsymbol{b})). \tag{66}$$

If (66) is solved directly via $\mathcal{P} = \boldsymbol{A}^{-1}$, then $\mathcal{P}\boldsymbol{r}(\boldsymbol{A}, \boldsymbol{b}) = \boldsymbol{A}^{-1}\boldsymbol{b} - \boldsymbol{x}$. So (66) then becomes $\mathcal{K}_{\boldsymbol{x}}(\boldsymbol{r}(\boldsymbol{I}, \boldsymbol{A}^{-1}\boldsymbol{b}))$, producing the exact solution of (64) in a single iteration of $\mathcal{K}$. Having established notation, we now present our implementation of static condensation via Slate by defining the appropriate operator, $\mathcal{P}$.

### 4.1  Interfacing with PETSc via custom preconditioners

The implementation of preconditioners for the systems considered in this paper requires manipulation not of assembled matrices, but rather their symbolic representation. To do this, we use the preconditioning infrastructure developed by Kirby and

Mitchell (2018), which gives preconditioners written in Python access to the symbolic problem description. In Firedrake, this means all derived preconditioners have direct access to the UFL representation of the PDE system. From this mathematical specification, we manipulate this appropriately via Slate and provide operators assembled from Slate expressions to PETSc for further algebraic preconditioning. Using this approach, we have developed a static condensation interface for the hybridization of $H(\text{div}) \times L^2$ mixed problems, and a generic interface for statically condensing finite element systems. The advantage of writing even the latter as a preconditioner is the ability to switch out the solution scheme for the system, even when nested inside a larger set of coupled equations or nonlinear solver (Newton-based methods) at *runtime*.

### 4.1.1 A static condensation interface for hybridization

As discussed in sections 3.1 and 3.2, one of the main advantages of using a hybridizable variant of a DG or mixed method is that such systems permit the use of cell-wise static condensation. To facilitate this, we provide a PETSc PC static condensation interface: `firedrake.SCPC`. This preconditioner takes the discretized system as in (44), and performs the local elimination and recovery procedures. Slate expressions are generated from the underlying UFL problem description.

More precisely, the incoming system has the form:

$$\begin{bmatrix} \boldsymbol{A}_{e,e} & \boldsymbol{A}_{e,c} \\ \boldsymbol{A}_{c,e} & \boldsymbol{A}_{c,c} \end{bmatrix} \begin{Bmatrix} \boldsymbol{X}_e \\ \boldsymbol{X}_c \end{Bmatrix} = \begin{Bmatrix} \boldsymbol{R}_e \\ \boldsymbol{R}_c \end{Bmatrix}, \tag{67}$$

where $\boldsymbol{X}_e$ is the vector of unknowns to be eliminated, $\boldsymbol{X}_c$ is the vector of unknowns for the condensed field, and $\boldsymbol{R}_e$, $\boldsymbol{R}_c$ are the incoming right-hand sides. The partitioning in (67) is determined by the solver option: `pc_sc_eliminate_fields`. Field indices are provided in the same way one configures solver options to PETSc. These indices determine which field(s) to statically condense into. For example, on a three-field problem (with indices 0, 1, and 2), setting `-pc_sc_eliminate_fields 0,1` will configure `firedrake.SCPC` to cell-wise eliminate fields 0 and 1; the resulting condensed system is associated with field 2.

The `firedrake.SCPC` preconditioner can be interpreted as a Schur-complement method for (67) of the form:

$$\mathcal{P} = \begin{bmatrix} \boldsymbol{I} & -\boldsymbol{A}_{e,e}^{-1}\boldsymbol{A}_{e,c} \\ \boldsymbol{0} & \boldsymbol{I} \end{bmatrix} \begin{bmatrix} \boldsymbol{A}_{e,e}^{-1} & \boldsymbol{0} \\ \boldsymbol{0} & \boldsymbol{S}^{-1} \end{bmatrix} \begin{bmatrix} \boldsymbol{I} & \boldsymbol{0} \\ -\boldsymbol{A}_{c,e}\boldsymbol{A}_{e,e}^{-1} & \boldsymbol{I} \end{bmatrix}, \tag{68}$$

where $\boldsymbol{S} = \boldsymbol{A}_{c,c} - \boldsymbol{A}_{c,e}\boldsymbol{A}_{e,e}^{-1}\boldsymbol{A}_{e,c}$ is the Schur-complement operator for the $\boldsymbol{X}_c$ system. The distinction here from block preconditioners via the PETSc `fieldsplit` option (Brown et al., 2012), for example, is that $\mathcal{P}$ does not require global actions; by design $\boldsymbol{A}_{e,e}^{-1}$ can be inverted locally and $\boldsymbol{S}$ is sparse. As a result, $\boldsymbol{S}$ can be assembled or applied *exactly*, up to numerical round-off, via Slate-generated kernels.

In practice, the only globally coupled system requiring iterative inversion is $\boldsymbol{S}$:

$$\mathcal{K}_{\boldsymbol{X}_c}(\mathcal{P}_1 r(\boldsymbol{S}, \boldsymbol{R}_s)), \tag{69}$$

where $\boldsymbol{R}_s = \boldsymbol{R}_c - \boldsymbol{A}_{c,e}\boldsymbol{A}_{e,e}^{-1}\boldsymbol{R}_e$ is the condensed right-hand side and $\mathcal{P}_1$ is a preconditioner for $\boldsymbol{S}$. Once $\boldsymbol{X}_c$ is computed, $\boldsymbol{X}_e$ is reconstructed by inverting the system $\boldsymbol{X}_e = \boldsymbol{A}_{e,e}^{-1}(\boldsymbol{R}_c - \boldsymbol{A}_{e,c}\boldsymbol{X}_c)$ cell-wise.

By construction, this preconditioner is suitable for both hybridized mixed and HDG discretizations. It can also be used within other contexts, such as the static condensation of continuous Galerkin discretizations (Guyan, 1965; Irons, 1965) or primal-hybrid methods (Devloo et al., 2018). As with any PETSc preconditioner, solver options can be specified for inverting $\boldsymbol{S}$ via the appropriate options prefix (`condensed_field`). The resulting `KSP` for (69) is compatible with existing solvers and external packages provided through the PETSc library. This allows users to experiment with a direct method and then switch to a more parallel-efficient iterative solver without changing the core application code.

### 4.1.2 Preconditioning mixed methods via hybridization

The preconditioner `firedrake.HybridizationPC` expands on the previous one, this time taking an $H(\text{div}) \times L^2$ system and automatically forming the hybridizable problem. This is accomplished through manipulating the UFL objects representing the discretized PDE. This includes replacing argument spaces with their discontinuous counterparts, introducing test functions on an appropriate trace space, and providing operators assembled from Slate expressions in a similar manner as described in Section 4.1.1.

More precisely, let $\boldsymbol{AX} = \boldsymbol{R}$ be the incoming mixed saddle point problem, where $\boldsymbol{R} = \left\{ \boldsymbol{R}_U \quad \boldsymbol{R}_P \right\}^T$, $\boldsymbol{X} = \left\{ \boldsymbol{U} \quad \boldsymbol{P} \right\}^T$, and $\boldsymbol{U}$ and $\boldsymbol{P}$ are the velocity and scalar unknowns respectively. Then this preconditioner replaces $\boldsymbol{AX} = \boldsymbol{R}$ with the extended problem:

$$
\begin{bmatrix} \widehat{\boldsymbol{A}} & \boldsymbol{C}^T \\ \boldsymbol{C} & \boldsymbol{0} \end{bmatrix} \left\{ \begin{matrix} \widehat{\boldsymbol{X}} \\ \boldsymbol{\Lambda} \end{matrix} \right\} = \left\{ \begin{matrix} \widehat{\boldsymbol{R}} \\ \boldsymbol{R}_g \end{matrix} \right\}
\tag{70}
$$

where $\boldsymbol{\Lambda}$ are the Lagrange multipliers, $\widehat{\boldsymbol{R}} = \left\{ \widehat{\boldsymbol{R}}_U \quad \boldsymbol{R}_P \right\}^T$, $\widehat{\boldsymbol{R}}_U$, $\boldsymbol{R}_P$ are the right-hand sides for the flux and scalar equations respectively, and $\widehat{\cdot}$ indicates modified matrices and co-vectors with discontinuous functions. Here, $\widehat{\boldsymbol{X}} = \left\{ \boldsymbol{U}^d \quad \boldsymbol{P} \right\}^T$ are the hybridizable (discontinuous) unknowns to be determined, and $\boldsymbol{C}\widehat{\boldsymbol{X}} = \boldsymbol{R}_g$ is the matrix representation of the transmission condition for the hybridizable mixed method (see (43)).

The application of `firedrake.HybridizationPC` can be interpreted as the Schur-complement reduction of (70):

$$
\widehat{\mathcal{P}} = \begin{bmatrix} \boldsymbol{I} & -\widehat{\boldsymbol{A}}^{-1}\boldsymbol{C}^T \\ \boldsymbol{0} & \boldsymbol{I} \end{bmatrix} \begin{bmatrix} \widehat{\boldsymbol{A}}^{-1} & \boldsymbol{0} \\ \boldsymbol{0} & \boldsymbol{S}^{-1} \end{bmatrix} \begin{bmatrix} \boldsymbol{I} & \boldsymbol{0} \\ -\boldsymbol{C}\widehat{\boldsymbol{A}}^{-1} & \boldsymbol{I} \end{bmatrix},
\tag{71}
$$

where $\boldsymbol{S}$ is the Schur-complement matrix $\boldsymbol{S} = -\boldsymbol{C}\widehat{\boldsymbol{A}}^{-1}\boldsymbol{C}^T$. As before, a single global system for $\boldsymbol{\Lambda}$ can be assembled cell-wise using Slate-generated kernels. Configuring the solver for inverting $\boldsymbol{S}$ is done via the PETSc options prefix: `-hybridization`. The recovery of $\boldsymbol{U}^d$ and $\boldsymbol{P}$ happens in the same manner as `firedrake.SCPC`.

Since the hybridizable flux solution is constructed in the "broken" $H(\text{div})$ space $U_h^d$, we must project the computed solution into $U_h \subset H(\text{div})$. This can be done cheaply via local facet averaging. The resulting solution is then updated via $\boldsymbol{U} \leftarrow \Pi_{\text{div}}\boldsymbol{U}^d$, where $\Pi_{\text{div}} : U_h^d \rightarrow U_h$ is a projection operator. This ensures the residual for the original mixed problem is properly evaluated to test for solver convergence. With $\widehat{\mathcal{P}}$ as in (71), the preconditioning operator for the original system $\boldsymbol{AX} = \boldsymbol{R}$ then has the

form:

$$\mathcal{P} = \mathbf{\Pi}\widehat{\mathcal{P}}\mathbf{\Pi}^T, \quad \mathbf{\Pi} = \begin{bmatrix} \Pi_{\text{div}} & \mathbf{0} & \mathbf{0} \\ \mathbf{0} & \boldsymbol{I} & \mathbf{0} \end{bmatrix}. \tag{72}$$

We note here that assembly of the right-hand side for the $\mathbf{\Lambda}$ system requires special attention. Firstly, when Neumann conditions are present, then $\boldsymbol{R}_g$ is not necessarily $\mathbf{0}$. Since the hybridization preconditioner has access to the entire Python context (which includes a list of boundary conditions and the spaces in which they are applied), surface integrals on the exterior boundary are added where appropriate and incorporated in the generated Slate expressions. A more subtle issue that requires extra care is the incoming right-hand side tested in the $H(\text{div})$ space $U_h$.

The situation we are given is that we have $\boldsymbol{R}_U = \boldsymbol{R}_U(\boldsymbol{w})$ for $\boldsymbol{w} \in U_h$, but require $\widehat{\boldsymbol{R}}_U(\boldsymbol{w}^d)$ for $\boldsymbol{w}^d \in U_h^d$. For consistency, we also require for any $\boldsymbol{w} \in U_h$ that

$$\widehat{\boldsymbol{R}}_U(\boldsymbol{w}) = \boldsymbol{R}_U(\boldsymbol{w}). \tag{73}$$

We can construct such a $\widehat{\boldsymbol{R}}_U$ satisfying (73) in the following way. By construction, we have for each basis function $\boldsymbol{\Psi}_i \in U_h$:

$$\boldsymbol{\Psi}_i = \begin{cases} \boldsymbol{\Psi}_i^d & \boldsymbol{\Psi}_i \text{ associated with an exterior facet node,} \\ \boldsymbol{\Psi}_i^{d,+} + \boldsymbol{\Psi}_i^{d,-} & \boldsymbol{\Psi}_i \text{ associated with an interior facet node,} \\ \boldsymbol{\Psi}_i^d & \boldsymbol{\Psi}_i \text{ associated with a cell interior node,} \end{cases} \tag{74}$$

where $\boldsymbol{\Psi}_i^d, \boldsymbol{\Psi}_i^{d,\pm} \in U_h^d$, and $\boldsymbol{\Psi}_i^{d,\pm}$ are basis functions corresponding to the positive and negative restrictions associated with the $i$-th facet node.[4] We then define our "broken" right-hand side via the local definition:

$$\widehat{\boldsymbol{R}}_U(\boldsymbol{\Psi}_i^d) = \frac{\boldsymbol{R}_U(\boldsymbol{\Psi}_i)}{N_i}, \tag{75}$$

where $N_i$ is the number of cells that the degree of freedom corresponding to $\boldsymbol{\Psi}_i \in U_h$ is topologically associated with. Using (74), (75), and the fact that $\boldsymbol{R}_U$ is linear in its argument, we can verify that our construction of $\widehat{\boldsymbol{R}}_U$ satisfies (73).

## 5 Numerical studies

We now present results utilizing the Slate DSL and our static condensation preconditioners for a set of test problems. Since we are using the interfaces outlined in Section 4, Slate is accessed indirectly and requires no manually-written solver code for hybridization or static condensation/local recovery. All parallel results were obtained on a single fully-loaded compute node of dual-socket Intel E5-2630v4 (Xeon) processors with $2 \times 10$ cores (2 threads per core) running at 2.2GHz. In order to avoid potential memory effects due to the operating system migrating processes between sockets, we pin MPI processes to cores.

Verification of the generated code is performed using parameter-sensitive convergence tests. The study consists of running a variety of discretizations spanning the methods outlined in Section 3. Details and numerical results are made public and can be viewed in Zenodo/Tabula-Rasa (2019) (see "Code and data availability"). All results are in full agreement with the theory.

---

[4] These are the two "broken" parts of $\boldsymbol{\Psi}_i$ on a particular facet connecting two elements. That is, for two adjacent cells, a basis function in $\boldsymbol{U}_h$ for a particular facet node can be decomposed into two basis functions in $U_h^d$ defined on their respective sides of the facet.

## 5.1 HDG method for a three-dimensional elliptic equation

In this section, we take a closer look at the LDG-H method for the model elliptic equation (sign-definite Helmholtz):

$$-\nabla \cdot \nabla p + p = f, \quad \text{in } \Omega = [0,1]^3, \tag{76}$$

$$p = g, \quad \text{on } \partial\Omega, \tag{77}$$

where $f$ and $g$ are chosen such that the analytic solution is $p = \exp\{\sin(\pi x)\sin(\pi y)\sin(\pi z)\}$. We use a regular mesh consisting $6 \cdot N^3$ tetrahedral elements ($N \in \{4, 8, 16, 32, 64\}$). First, we reformulate (76)–(77) as the mixed problem:

$$\boldsymbol{u} + \nabla p = 0, \tag{78}$$

$$\nabla \cdot \boldsymbol{u} + p = f, \tag{79}$$

$$p = g, \quad \text{on } \partial\Omega. \tag{80}$$

We start with linear polynomial approximations, up to cubic, for the LDG-H discretization of (78)–(80). Additionally, we compute a post-processed scalar approximation $p_h^\star$ of the HDG solution. This raises the approximation order of the computed solution by an additional degree. In all numerical studies here, we set the HDG parameter $\tau = 1$. All results were computed in parallel, utilizing a single compute node (described previously).

A continuous Galerkin (CG) discretization of the primal problem (76)–(77) serves as a reference for this experiment. Due to
the superconvergence in the post-processed solution for the HDG method, we use CG discretizations of polynomial order 2, 3, and 4. This takes into account the enhanced accuracy of the HDG solution, despite being initially computed as a lower-order approximation. We therefore expect both methods to produce equally accurate solutions to the model problem.

Our aim here is not to compare the performance of HDG and CG, which has been investigated elsewhere (for example, see Kirby et al. (2012); Yakovlev et al. (2016)). Instead, we provide a reference that the reader might be more familiar with in order
to evaluate whether our software framework produces a sufficiently performant HDG implementation relative to what might be expected.

To invert the CG system, we use a conjugate gradient solver with hypre's boomerAMG implementation of algebraic multigrid (AMG) as a preconditioner (Falgout et al., 2006). For the HDG method, we use the preconditioner described in Section 4.1.1 and the same solver setup as the CG method for the trace system. While indeed the trace operator is symmetric and positive-
definite, one should keep in mind that conclusions regarding the performance of off-the-shelf AMG packages on the HDG trace system is still relatively unclear. As a result, efforts on developing more efficient multigrid strategies is a topic of on-going interest (Cockburn et al., 2014; Kronbichler and Wall, 2018).

To avoid over-solving, we iterate to a relative tolerance such that the *discretization error* is minimal for a given mesh. In other words, the solvers are configured to terminate when there is no further reduction in the $L^2$-error of the computed solution
compared with the analytic solution. This means we are not iterating to a fixed solver tolerance across all mesh resolutions. Therefore, we can expect the total number of Krylov iterations (for both the CG and HDG methods) to increase as the mesh resolution becomes finer. The rationale behind this approach is to directly compare the execution time to solve for the best possible approximation to the solution given a fixed resolution.

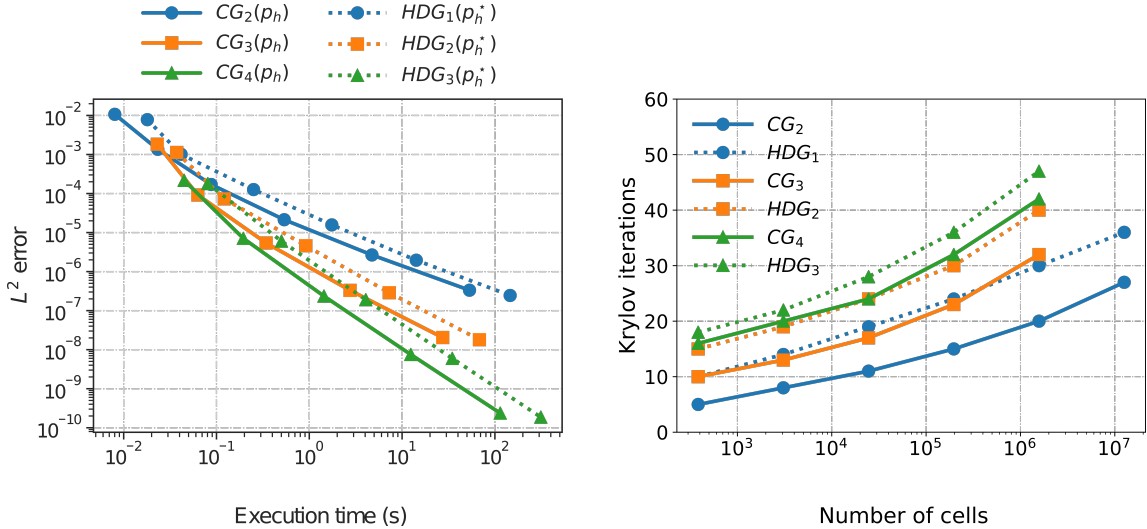

**Figure 4.** Comparison of continuous Galerkin and LDG-H solvers for the model three-dimensional positive-definite Helmholtz equation. **Left**: a log-log plot showing the error against execution time for the CG and HDG with post-processing ($\tau = 1$) methods. **Right**: A log-linear plot showing Krylov iterations of the AMG-preconditioned conjugate gradient algorithm (to reach discretization error) against number of cells.

### 5.1.1 Error versus execution time

The total execution time is recorded for the CG and HDG solvers, which includes the setup time for the AMG preconditioner, matrix assembly, and the time-to-solution for the Krylov method. In the HDG case, we include all setup costs, the time spent building the Schur-complement for the traces, local recovery of the scalar and flux approximations, and post-processing. The
$L^2$-error against execution time and Krylov iterations to reach discretization error for each mesh are summarized in Figure 4.

The HDG method of order $k-1$ ($HDG_{k-1}$) with post-processing, as expected, produces a solution which is as accurate as the CG method of order $k$ ($CG_k$). While the full HDG system is never explicitly assembled, the larger execution time is a result of several factors. The primary factor is that the total number of trace unknowns for the $HDG_1$, $HDG_2$, and $HDG_3$ discretizations is roughly four, three, and two times larger (resp.) than the corresponding number of CG unknowns. Therefore,
each iteration is more expensive. We also observe that the trace system requires more Krylov iterations to reach discretization error, which appears to improve relative to the CG method as the approximation order increases. Further analysis on multigrid methods for HDG systems is required to draw further conclusions. The main computational bottleneck in HDG methods is the global linear solver. We therefore expect our implementation to be dominated by the cost associated with inverting the trace operator. If one considers just the time-to-solution, the CG method is clearly ahead of the HDG method. However, the
superior scaling, local conservation, and stabilization properties of the HDG method make it a particularly appealing choice for fluid dynamics applications (Yakovlev et al., 2016; Kronbichler and Wall, 2018). Therefore, the development of good preconditioning strategies for the HDG method is critical for its competitive use.

### 5.1.2 Break down of solver time

The HDG method requires many more degrees of freedom than CG or primal DG methods. This is largely due to the fact that the HDG method simultaneously approximates the primal solution and its velocity. The global matrix for the traces is significantly larger than the one for the CG system at low polynomial order. The execution time for HDG is then compounded by a more expensive global solve.

Figure 5 displays a break down of total execution times on a simplicial mesh consisting of $1.5$ million elements. The execution times have been normalized by the CG total time, showing that the HDG method is roughly 3 times more expensive than the CG method. This is expected given the larger degree-of-freedom count and expensive global solve. The raw numerical breakdown of the HDG and CG solvers are shown in Table 1. We isolate each component of the HDG method contributing to the total execution time. Local operations include static condensation (trace operator assembly), forward elimination (right-hand side assembly for the trace system), backwards substitution to recover the scalar and velocity unknowns, and local post-processing of the scalar solution. For all $k$, our HDG implementation is solver-dominated as expected.

Both trace operator and right-hand side assembly are dominated by the costs of inverting a local square mixed matrix coupling the scalar and velocity unknowns, which is performed directly via an LU factorization. This is also the case for backwards substitution. They should all therefore be of the same magnitude in time spent. We observe that this is the case across all degrees, with times ranging between approximately 6—11% of total execution time for assembling the condensed system.

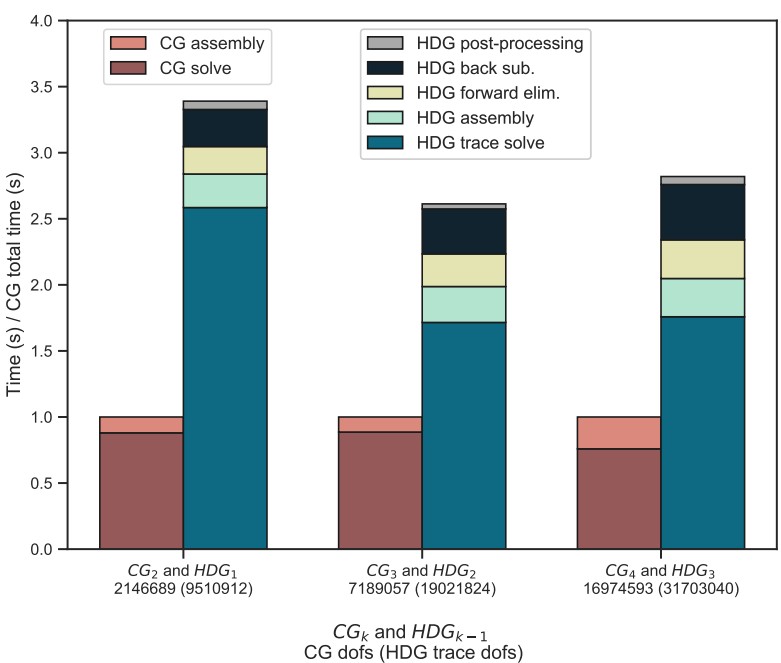

**Figure 5.** Break down of the $CG_k$ and $HDG_{k-1}$ execution times on a $6 \cdot 64^3$ simplicial mesh.

**Table 1.** Breakdown of the raw timings for the $HDG_{k-1}$ ($\tau = 1$) and $CG_k$ methods, $k = 2$, 3, and 4. Each method corresponds to a mesh size $N = 64$ on a fully-loaded compute node.

| Stage | $HDG_1$ | | $HDG_2$ | | $HDG_3$ | |
|---|---|---|---|---|---|---|
| | $t_{\text{stage}}$ (s) | $\% \, t_{\text{total}}$ | $t_{\text{stage}}$ (s) | $\% \, t_{\text{total}}$ | $t_{\text{stage}}$ (s) | $\% \, t_{\text{total}}$ |
| Matrix assembly (static cond.) | 1.05 | 7.49 % | 6.95 | 10.40 % | 31.66 | 10.27 % |
| Forward elimination | 0.86 | 6.13 % | 6.32 | 9.45 % | 31.98 | 10.37 % |
| Trace solve | 10.66 | 76.24 % | 43.89 | 65.66 % | 192.31 | 62.36 % |
| Back substitution | 1.16 | 8.28 % | 8.71 | 13.03 % | 45.81 | 14.85 % |
| Post processing | 0.26 | 1.86 % | 0.98 | 1.46 % | 6.62 | 2.15 % |
| HDG Total | 13.98 | | 66.85 | | 308.37 | |
| | $CG_2$ | | $CG_3$ | | $CG_4$ | |
| | $t_{\text{stage}}$ (s) | $\% \, t_{\text{total}}$ | $t_{\text{stage}}$ (s) | $\% \, t_{\text{total}}$ | $t_{\text{stage}}$ (s) | $\% \, t_{\text{total}}$ |
| Matrix assembly (monolithic) | 0.50 | 12.01 % | 2.91 | 11.39 % | 26.37 | 24.11 % |
| Solve | 3.63 | 87.99 % | 22.67 | 88.61 % | 82.99 | 75.89 % |
| CG Total | 4.12 | | 25.59 | | 109.36 | |

Back-substitution takes roughly the same time as the static condensation and forward elimination stages (approximately 12% of execution time on average). Finally, the additional cost of post-processing accrues negligible time (roughly 2% of execution time across all degrees). This is a small cost for an increase in order of accuracy.

We note that caching of local tensors does not occur. Each pass to perform the local eliminations and backwards reconstruc-
5  tions rebuilds the local element tensors. It is not clear at this time whether the performance gained from avoiding rebuilding the local operators will offset the memory costs of storing the local matrices. Moreover, in time-dependent problems where the operators may contain state-dependent variables, rebuilding local matrices will be necessary in each time-step regardless.

## 5.2 Hybridizable mixed methods for the shallow water equations

A primary motivator for our interest in hybridizable methods revolves around developing efficient solvers for problems in
10  geophysical flows. In this section, we present some resulting integrating the nonlinear, rotating shallow water equations on the sphere using test case 5 (flow past an isolated mountain) from Williamson et al. (1992). For our discretization approach, we use the framework of compatible finite elements (Cotter and Shipton, 2012; Cotter and Thuburn, 2014).

The model equations we consider are the vector-invariant rotating nonlinear shallow water system defined on a two-dimensional spherical surface $\Omega$ embedded in $\mathbb{R}^3$:

$$\frac{\partial \boldsymbol{u}}{\partial t} + \left( \nabla^\perp \cdot \boldsymbol{u} + f \right) \boldsymbol{u}^\perp + \nabla \left( g \left( D + b \right) + \frac{1}{2} |\boldsymbol{u}|^2 \right) = 0, \tag{81}$$

$$\frac{\partial D}{\partial t} + \nabla \cdot \left( \boldsymbol{u} D \right) = 0, \tag{82}$$

where $\boldsymbol{u}$ is the fluid velocity, $D$ is the depth field, $f$ is the Coriolis parameter, $g$ is the acceleration due to gravity, $b$ is the bottom topography, and $(\cdot)^\perp \equiv \hat{\boldsymbol{z}} \times \cdot$, with $\hat{\boldsymbol{z}}$ being the unit normal to the surface $\Omega$. After discretizing in time and space using a semi-implicit scheme and Picard linearization, following Natale et al. (2016), we must solve a sequence of saddle point system at each time-step of the form:

$$
\qquad \begin{bmatrix} \boldsymbol{A} & -g\frac{\Delta t}{2}\boldsymbol{B}^T \\ H\frac{\Delta t}{2}\boldsymbol{B} & \boldsymbol{M} \end{bmatrix} \begin{Bmatrix} \Delta U \\ \Delta D \end{Bmatrix} = \begin{Bmatrix} \boldsymbol{R_u} \\ \boldsymbol{R_D} \end{Bmatrix}. \qquad\qquad (83)
$$

See Appendix A for a complete description of the entire discretization strategy. The system (83) is the matrix equation corresponding to the linearized equations in (A3)–(A4).

The Picard updates $\Delta U$ and $\Delta D$ are sought in the mixed finite element spaces $U_h \subset H(\mathrm{div})$ and $V_h \subset L^2$ respectively. Stable mixed finite element pairings correspond to the well-known RT and BDM mixed methods, such as: $\mathrm{RT}_k \times \mathrm{DG}_{k-1}$ or

$\mathrm{BDM}_k \times \mathrm{DG}_{k-1}$. These also fall within the set of compatible mixed spaces ideal for geophysical fluid dynamics (Cotter and Shipton, 2012; Natale et al., 2016; Melvin et al., 2019). In particular, the lowest-order RT method ($\mathrm{RT}_1 \times \mathrm{DG}_0$) on a structured quadrilateral grid (such as the latitude-longitude grid used many operational dynamical cores) is analogous to the Arakawa C-grid finite difference discretization.

In staggered finite difference models, the standard approach for solving (83) is to neglect the Coriolis term and eliminate the

velocity unknown $\Delta U$ to obtain a discrete elliptic equation for $\Delta D$, where smoothers like Richardson iterations or relaxation methods are convergent. This is more problematic in the compatible finite element framework, since $\boldsymbol{A}$ has a dense inverse. Instead, we use the preconditioner described in Section 4.1.2 to form the equivalent hybridizable formulation, where both $\Delta U$ and $\Delta D$ are eliminated locally to produce a sparse elliptic equation for the Lagrange multipliers.

### 5.2.1 Atmospheric flow over a mountain

As a test problem, we solve test case 5 of Williamson et al. (1992), on the surface of a sphere with radius $R = 6371\mathrm{km}$. We refer the reader to Cotter and Shipton (2012); Shipton et al. (2018) for a more comprehensive study on mixed finite elements for shallow water systems of this type. We use the mixed finite element pairs $(\mathrm{RT}_1, \mathrm{DG}_0)$ (lowest-order RT method) and $(\mathrm{BDM}_2, \mathrm{DG}_1)$ (next-to-lowest order BDM method) for the velocity and depth spaces. A mesh of the sphere is generated from 7 refinements of an icosahedron, resulting in a triangulation $\mathcal{T}_h$ consisting of 327,680 elements in total. The grid information

for both mixed methods are summarized in Table 2.

We run for a total of 25 time-steps, with a fixed number of 4 Picard iterations in each time-step. We compare the overall simulation time using two different solver configurations for the implicit linear system. First, we use a flexible variant of GMRES [5] acting on the system (83) with an approximate Schur complement preconditioner:

$$
\mathcal{P}_{\mathrm{SC}} = \begin{bmatrix} \boldsymbol{I} & g\frac{\Delta t}{2}\boldsymbol{A}^{-1}\boldsymbol{B}^T \\ \boldsymbol{0} & \boldsymbol{I} \end{bmatrix} \begin{bmatrix} \boldsymbol{A}^{-1} & \boldsymbol{0} \\ \boldsymbol{0} & \widetilde{\boldsymbol{S}}^{-1} \end{bmatrix} \begin{bmatrix} \boldsymbol{I} & \boldsymbol{0} \\ -H\frac{\Delta t}{2}\boldsymbol{B}\boldsymbol{A}^{-1} & \boldsymbol{I} \end{bmatrix}, \qquad\qquad (84)
$$

---

[5]We use a flexible version of GMRES on the outer system since we use an additional Krylov solver to iteratively invert the Schur-complement.

**Table 2.** The number of unknowns to be determined are summarized for each compatible finite element method. Resolution is the same for both methods.

| Discretization properties | | | | | |
|---|---|---|---|---|---|
| Mixed method | # cells | $\Delta x$ | Velocity unknowns | Depth unknowns | Total (millions) |
| $\mathrm{RT}_1 \times \mathrm{DG}_0$ | 327,680 | $\approx 43$ km | 491,520 | 327,680 | 0.8 M |
| $\mathrm{BDM}_2 \times \mathrm{DG}_1$ | | | 2,457,600 | 983,040 | 3.4 M |

where $\widetilde{\boldsymbol{S}} = \boldsymbol{M} + gH\frac{\Delta t^2}{4}\boldsymbol{B}\mathrm{diag}(\boldsymbol{A})^{-1}\boldsymbol{B}^T$, and $\mathrm{diag}(\boldsymbol{A})$ is a diagonal approximation to the velocity mass matrix (plus the addition of a Coriolis matrix). The Schur-complement system is inverted via GMRES due to the asymmetry from the Coriolis term, with the inverse of $\widetilde{\boldsymbol{S}}$ as the preconditioning operator. The sparse approximation $\widetilde{\boldsymbol{S}}$ is inverted using PETSc's smoothed aggregation multigrid (GAMG). The Krylov method is set to terminate once the preconditioned residual norm is reduced by a

5 factor of $10^8$. $\boldsymbol{A}^{-1}$ is computed approximately using a single application of incomplete LU (zero fill-in).

Next, we use only the application of our hybridization preconditioner (no outer Krylov method), which replaces the original linearized mixed system with its hybridizable equivalent. After hybridization, we have the following extended problem for the Picard updates: find $(\Delta \boldsymbol{u}_h^d, \Delta D_h, \lambda_h) \in U_h^d \times V_h \times M_h$ satisfying

$$\left(\boldsymbol{w}, \Delta \boldsymbol{u}_h^d\right)_{\mathcal{T}_h} + \frac{\Delta t}{2}\left(\boldsymbol{w}, f\left(\Delta \boldsymbol{u}_h^d\right)^{\perp}\right)_{\mathcal{T}_h} - \frac{\Delta t}{2}\left(\nabla \cdot \boldsymbol{w}, g\Delta D_h\right)_{\mathcal{T}_h} + \langle [[\boldsymbol{w}]], \lambda_h\rangle_{\partial \mathcal{T}_h} = \widehat{R}_{\boldsymbol{u}}, \quad \forall \boldsymbol{w} \in U_h^d, \tag{85}$$

$$\left(\phi, \Delta D_h\right)_{\mathcal{T}_h} + \frac{\Delta t}{2}\left(\phi, H\nabla \cdot \Delta \boldsymbol{u}_h^d\right)_{\mathcal{T}_h} = R_D, \quad \forall \phi \in V_h, \tag{86}$$

$$\left\langle \gamma, \left[\left[\Delta \boldsymbol{u}_h^d\right]\right]\right\rangle_{\partial \mathcal{T}_h} = 0, \quad \forall \gamma \in M_h. \tag{87}$$

Note that the space $M_h$ is chosen such that the trace functions, when restricted to a facet $e \in \partial \mathcal{T}_h$, are in the same polynomial space as $\Delta \boldsymbol{u}_h \cdot \boldsymbol{n}|_e$. Moreover, it can be shown that the Lagrange multiplier $\lambda_h$ is an approximation to the depth unknown $\Delta t g \Delta D / 2$ restricted to $\partial \mathcal{T}_h$.

15 The resulting three-field problem in (85)–(87) produces the following matrix equation:

$$\begin{bmatrix} \widehat{\boldsymbol{\mathcal{A}}} & \boldsymbol{C}^T \\ \boldsymbol{C} & \boldsymbol{0} \end{bmatrix} \begin{Bmatrix} \Delta \boldsymbol{X} \\ \boldsymbol{\Lambda} \end{Bmatrix} = \begin{Bmatrix} \widehat{\boldsymbol{R}}_{\Delta X} \\ \boldsymbol{0} \end{Bmatrix} \tag{88}$$

where $\widehat{\boldsymbol{\mathcal{A}}}$ is the discontinuous operator coupling $\Delta \boldsymbol{X} = \left\{ \Delta \boldsymbol{U}^d \quad \Delta \boldsymbol{D} \right\}^T$ and $\boldsymbol{R}_{\Delta X} = \left\{ \widehat{\boldsymbol{R}}_{\boldsymbol{u}} \quad \boldsymbol{R}_D \right\}^T$ are the problem residuals. An exact Schur-complement factorization is performed on (88), using Slate to generate the local elimination kernels. We use the same set of solver options for the inversion of $\widetilde{\boldsymbol{S}}$ in (84) to invert the Lagrange multiplier system. The increments

20 $\Delta \boldsymbol{U}^d$ and $\Delta \boldsymbol{D}$ are recovered locally, using Slate-generated kernels. Once recovery is complete, $\Delta \boldsymbol{U}^d$ is projected back into the conforming $H(\mathrm{div})$ finite element space via $\Delta \boldsymbol{U} \leftarrow \Pi_{\mathrm{div}}\Delta \boldsymbol{U}^d$. Based on the discussion in Section 4.1.2, we apply:

$$\mathcal{P}_{\mathrm{hybrid}} = \Pi \left( \begin{bmatrix} \boldsymbol{I} & -\widehat{\boldsymbol{\mathcal{A}}}^{-1}\boldsymbol{C}^T \\ \boldsymbol{0} & \boldsymbol{I} \end{bmatrix} \begin{bmatrix} \widehat{\boldsymbol{\mathcal{A}}}^{-1} & \boldsymbol{0} \\ \boldsymbol{0} & \boldsymbol{S}^{-1} \end{bmatrix} \begin{bmatrix} \boldsymbol{I} & \boldsymbol{0} \\ -\boldsymbol{C}\widehat{\boldsymbol{\mathcal{A}}}^{-1} & \boldsymbol{I} \end{bmatrix} \right) \Pi^T. \tag{89}$$

**Table 3.** Preconditioner solve times for a 25-step run with $\Delta t = 100$s. These are cumulative times in each stage of the two preconditioners throughout the entire profile run. We display the average iteration count (rounded to the nearest integer) for both the outer and the inner Krylov solvers. The significant speedup when using hybridization is a direct result of eliminating the outer-most solver.

| Preconditioner and solver details | | | | | |
|---|---|---|---|---|---|
| Mixed method | Preconditioner | $t_{\text{total}}$ (s) | Avg. outer its. | Avg. inner its. | $\frac{t_{\text{total}}^{\text{SC}}}{t_{\text{total}}^{\text{hybrid.}}}$ |
| RT$_1 \times$ DG$_0$ | approx. Schur. ($\mathcal{P}_{\text{SC}}$) | 15.137 | 2 | 8 | 3.413 |
| | hybridization ($\mathcal{P}_{\text{hybrid}}$) | 4.434 | None | 2 | |
| BDM$_2 \times$ DG$_1$ | approx. Schur. ($\mathcal{P}_{\text{SC}}$) | 300.101 | 4 | 9 | 5.556 |
| | hybridization ($\mathcal{P}_{\text{hybrid}}$) | 54.013 | None | 6 | |

**Table 4.** Breakdown of the cost (average) of a single application of the preconditioned flexible GMRES method and hybridization preconditioner. Hybridization takes approximately the same time per iteration.

| Preconditioner | Stage | RT$_1 \times$ DG$_0$ | | BDM$_2 \times$ DG$_1$ | |
|---|---|---|---|---|---|
| | | $t_{\text{stage}}$ (s) | % $t_{\text{total}}$ | $t_{\text{stage}}$ (s) | % $t_{\text{total}}$ |
| approx. Schur ($\mathcal{P}_{\text{SC}}$) | Schur solve | 0.07592 | 91.28 % | 0.78405 | 93.53 % |
| | invert velocity operator: $\boldsymbol{A}$ | 0.00032 | 0.39 % | 0.00678 | 0.81 % |
| | apply inverse: $\boldsymbol{A}^{-1}$ | 0.00041 | 0.49 % | 0.00703 | 0.84 % |
| | gmres other | 0.00652 | 7.84 % | 0.04041 | 4.82 % |
| | Total | 0.08317 | | 0.83827 | |
| hybridization ($\mathcal{P}_{\text{hybrid}}$) | Transfer: $\boldsymbol{R}_{\Delta X} \rightarrow \widehat{\boldsymbol{R}}_{\Delta X}$ | 0.00322 | 7.26 % | 0.00597 | 1.10 % |
| | Forward elim.: $-\boldsymbol{C}\widehat{\boldsymbol{A}}^{-1}\widehat{\boldsymbol{R}}_{\Delta X}$ | 0.00561 | 12.64 % | 0.12308 | 22.79 % |
| | Trace solve | 0.02289 | 51.63 % | 0.28336 | 52.46 % |
| | Back sub. | 0.00986 | 22.23 % | 0.12220 | 22.62 % |
| | Projection: $\Pi_{\text{div}}\Delta \boldsymbol{U}^d$ | 0.00264 | 5.96 % | 0.00516 | 0.96 % |
| | Total | 0.04434 | | 0.54013 | |

Table 3 displays a summary of our findings. The advantages of a hybridizable method versus a mixed method are more clearly realized in this experiment. When using hybridization, we observe a significant reduction in time spent in the implicit solver compared to the approximate Schur-complement approach. This is primarily because we have reduced the number of "outer" iterations to zero; the hybridization preconditioner is performing an *exact* factorization of the global hybridizable system. This is empirically supported when considering per-application solve times. The values reported in Table 4 show the average cost of a single outer GMRES iteration (which includes the application of $\mathcal{P}_{\text{SC}}$) and a single application of $\mathcal{P}_{\text{hybrid}}$. Hybridization and the approximate Schur complement preconditioner are comparable in terms of average execution time, with hybridization being slightly faster. This further demonstrates that the primary cause for the longer execution time of the latter is directly

related to the additional outer iterations induced from using an approximate factorization. In terms of over all time-to-solution, the hybridizable methods are clearly ahead of the original mixed methods.

We also measure the relative reductions in the problem residual of the linear system (83). Our hybridization preconditioner reduces the residual by a factor of $10^8$ on average, which coincides with the specified relative tolerance for the Krylov method on the trace system. In other words, the reduction in the residual for the trace system translates to an overall reduction in the residual for the mixed system by the same factor.

The test case was run up to day 15 on a coarser resolution (20,480 simplicial cells with $\Delta x \approx 210$km) and a time-step size $\Delta t = 500$ seconds. Snapshots of the entire simulation are provided in Figure 6 using the semi-implicit scheme described in Appendix A. The results we have obtained for days 5, 10, and 15 are comparable to the corresponding results of Nair et al. (2005); Ullrich et al. (2010); Kang et al. (2019). We refer the reader to Shipton et al. (2018) for further demonstrations of shallow water test cases featuring the use of the hybridization preconditioner described in Section 4.1.2.

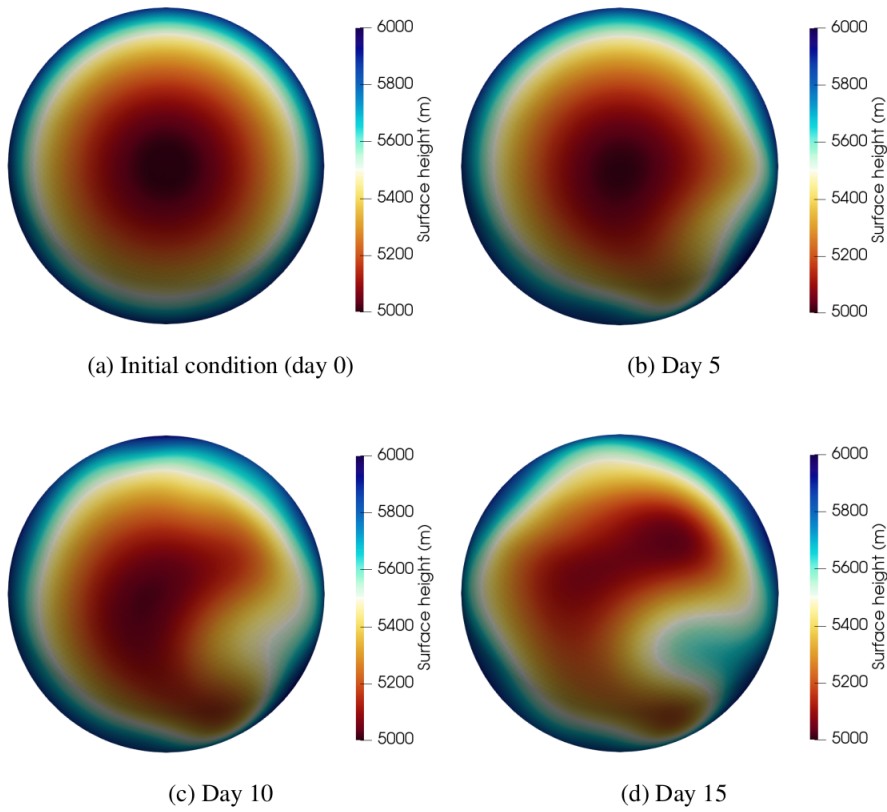

(a) Initial condition (day 0)   (b) Day 5

(c) Day 10   (d) Day 15

**Figure 6.** Snapshots (view from the northern pole) from the isolated mountain test case. The surface height (m) at days 5, 10, and 15. The snapshots were generated on a mesh with $20,480$ simplicial cells, a $\text{BDM}_2 \times \text{DG}_1$ discretization, and $\Delta t = 500$ seconds. The linear system during each Picard cycle was solved using the hybridization preconditioner.

## 5.3 Hybridizable methods for a linear Boussinesq model

As a final example, we consider the simplified atmospheric model obtained from a linearization of the compressible Boussinesq equations in a rotating domain:

$$\frac{\partial \boldsymbol{u}}{\partial t} + 2\boldsymbol{\Omega} \times \boldsymbol{u} = -\nabla p + b\hat{\boldsymbol{z}}, \tag{90}$$

$$\frac{\partial p}{\partial t} = -c^2 \nabla \cdot \boldsymbol{u}, \tag{91}$$

$$\frac{\partial b}{\partial t} = -N^2 \boldsymbol{u} \cdot \hat{\boldsymbol{z}}, \tag{92}$$

where $\boldsymbol{u}$ is the fluid velocity, $p$ the pressure, $b$ is the buoyancy, $\boldsymbol{\Omega}$ the planetary angular rotation vector, $c$ is the speed of sound ($\approx 343\mathrm{ms}^{-1}$), and $N$ is the buoyancy frequency ($\approx 0.01\mathrm{s}^{-1}$). Equations (90)–(92) permit fast-moving acoustic waves driven by perturbations in $b$. This is the model presented in Skamarock and Klemp (1994), which uses a quadratic equation of state to avoid some of the complications of the full compressible Euler equations (the hybridization of which we shall address in future work). We solve these equations subject to the rigid-lid condition $\boldsymbol{u} \cdot \boldsymbol{n} = 0$ on all boundaries.

Our domain consists of a spherical annulus, with the mesh constructed from a horizontal "base" mesh of the surface of a sphere of radius $R$, extruded upwards by a height $H_\Omega$. The vertical discretization is a structured one-dimensional grid, which facilitates the staggering of thermodynamic variables, such as $b$. We consider two kinds of meshes: one obtained by extruding an icosahedral sphere mesh, and another from a cubed sphere.

Since our mesh has a natural tensor product structure, we construct suitable finite element spaces constructed by taking the tensor product of a horizontal space with a vertical space. To ensure our discretization is "compatible," we use the one- and two-dimensional finite element de-Rham complexes: $V_h^0 \xrightarrow{\partial_{\hat{z}}} V_h^1$ and $U_h^0 \xrightarrow{\nabla^\perp} U_h^1 \xrightarrow{\nabla \cdot} U_h^2$. We can then construct the three-dimensional complex: $W_h^0 \xrightarrow{\nabla} W_h^1 \xrightarrow{\nabla \times} W_h^2 \xrightarrow{\nabla \cdot} W_h^3$, where

$$W_h^0 = U_h^0 \otimes V_h^0, \tag{93}$$

$$W_h^1 = \texttt{HCurl}(U_h^1 \otimes V_h^0) \oplus \texttt{HCurl}(U_h^0 \otimes V_h^1) =: W_h^{1,h} \oplus W_h^{1,v}, \tag{94}$$

$$W_h^2 = \texttt{HDiv}(U_h^1 \otimes V_h^1) \oplus \texttt{HDiv}(U_h^2 \otimes V_h^0) =: W_h^{2,h} \oplus W_h^{2,v}, \tag{95}$$

$$W_h^3 = U_h^2 \otimes V_h^1. \tag{96}$$

Here, $\texttt{HCurl}$ and $\texttt{HDiv}$ denote operators which ensure the correct Piola transformations are applied when mapping from physical to reference element. We refer the reader to McRae et al. (2016) for an overview of constructing tensor product finite element spaces in Firedrake. For the analysis of compatible finite element discretizations and their relation to the complex (93)–(96), we refer the reader to Natale et al. (2016). Each discretization used in this section is constructed from more familiar finite element families, shown in Table 5.

**Table 5.** Vertical and horizontal spaces for the three-dimensional compatible finite element discretization of the linear Boussinesq model. The $RT_k$ and $BDFM_{k+1}$ methods are constructed on triangular prism elements, while the $RTCF_k$ method is defined on extruded quadrilateral elements.

| | Compatible finite element spaces | | | | |
|---|---|---|---|---|---|
| Mixed method | $V_h^0$ | $V_h^1$ | $U_h^0$ | $U_h^1$ | $U_h^2$ |
| $RT_k$ | $CG_k([0, H_\Omega])$ | $DG_{k-1}([0, H_\Omega])$ | $CG_k(\triangle)$ | $RT_k(\triangle)$ | $DG_{k-1}(\triangle)$ |
| $BDFM_{k+1}$ | $CG_{k+1}([0, H_\Omega])$ | $DG_k([0, H_\Omega])$ | $CG_{k+1}(\triangle)$ | $BDFM_{k+1}(\triangle)$ | $DG_k(\triangle)$ |
| $RTCF_k$ | $CG_k([0, H_\Omega])$ | $DG_{k-1}([0, H_\Omega])$ | $Q_k(\square)$ | $RTCF_k(\square)$ | $DQ_{k-1}(\square)$ |

### 5.3.1 Compatible finite element discretization

A compatible finite element discretization of (90)–(92) constructs solutions in the following finite element spaces:

$$\boldsymbol{u}_h \in \mathring{W}_h^2, \quad p_h \in W_h^3, \quad b_h \in W_h^b, \tag{97}$$

where $\mathring{W}_h^2$ is the subspace of $W_h^2 \subset H(\mathrm{div})$ whose functions $\boldsymbol{w}$ satisfy $\boldsymbol{w} \cdot \boldsymbol{n} = 0$ on $\partial \Omega$, $W_h^3 \subset L^2$, and $W_h^b \equiv U_h^2 \otimes V_h^0$. Note that $W_h^b$ is just the scalar version of the vertical velocity space.[6] That is, $W_h^b$ and $W_h^{2,v}$ have the same number of degrees of freedom, but differ in how they are pulled back to the reference element.

To obtain the discrete system, we simply multiply equations (90)–(92) by test functions $\boldsymbol{w} \in \mathring{W}_h^2$, $\phi \in W_h^3$ and $\eta \in W_h^b$ and integrate by parts. We introduce the increments $\delta \boldsymbol{u}_h \equiv \boldsymbol{u}_h^{n+1} - \boldsymbol{u}_h^n$, and set $\boldsymbol{u}_0 \equiv \boldsymbol{u}_h^n$ (similarly for $\delta p_h$, $p_0$, $\delta b_h$, and $b_0$). Using an implicit midpoint rule discretization, we need to solve the following mixed problem at each time-step: find $\delta \boldsymbol{u}_h \in \mathring{W}_h^2$, $\delta p_h \in W_h^3$ and $\delta b_h \in W_h^b$ such that

$$(\boldsymbol{w}, \delta \boldsymbol{u}_h)_{\mathcal{T}_h} + \frac{\Delta t}{2}(\boldsymbol{w}, 2\boldsymbol{\Omega} \times \delta \boldsymbol{u}_h)_{\mathcal{T}_h} - \frac{\Delta t}{2}(\nabla \cdot \boldsymbol{w}, \delta p_h)_{\mathcal{T}_h} - \frac{\Delta t}{2}(\boldsymbol{w}, \delta b_h \hat{\boldsymbol{z}})_{\mathcal{T}_h} = r_u, \quad \forall \boldsymbol{w} \in \mathring{W}_h^2 \tag{98}$$

$$(\phi, \delta p_h)_{\mathcal{T}_h} + \frac{\Delta t}{2}c^2 (\phi, \nabla \cdot \delta \boldsymbol{u}_h)_{\mathcal{T}_h} = r_p, \quad \forall \phi \in W_h^3, \tag{99}$$

$$(\eta, \delta b_h)_{\mathcal{T}_h} + \frac{\Delta t}{2}N^2 (\eta, \delta \boldsymbol{u}_h \cdot \hat{\boldsymbol{z}})_{\mathcal{T}_h} = r_b, \quad \forall \eta \in W_h^b, \tag{100}$$

where the residuals are: $r_u = -\Delta t (\boldsymbol{w}, 2\boldsymbol{\Omega} \times \boldsymbol{u}_0)_{\mathcal{T}_h}$, $r_p = -c^2 \Delta t (\phi, \nabla \cdot \boldsymbol{u}_0)_{\mathcal{T}_h}$, and $r_b = -N^2 \Delta t (\eta, \boldsymbol{u}_0 \cdot \hat{\boldsymbol{z}})_{\mathcal{T}_h}$.

The resulting matrix equations have the form:

$$\begin{bmatrix} \boldsymbol{A_u} & -\frac{\Delta t}{2}\boldsymbol{D}^T & -\frac{\Delta t}{2}\boldsymbol{Q}^T \\ \frac{\Delta t}{2}c^2\boldsymbol{D} & \boldsymbol{M}_p & \boldsymbol{0} \\ \frac{\Delta t}{2}N^2\boldsymbol{Q} & \boldsymbol{0} & \boldsymbol{M}_b \end{bmatrix} \begin{Bmatrix} \boldsymbol{U} \\ \boldsymbol{P} \\ \boldsymbol{B} \end{Bmatrix} = \begin{Bmatrix} \boldsymbol{R_u} \\ \boldsymbol{R}_p \\ \boldsymbol{R}_b \end{Bmatrix}, \tag{101}$$

where $\boldsymbol{A_u} = \boldsymbol{M_u} + \frac{\Delta t}{2}\boldsymbol{C_\Omega}$, $\boldsymbol{C_\Omega}$ is the asymmetric matrix associated with the Coriolis term, $\boldsymbol{M_u}$, $\boldsymbol{M}_p$, $\boldsymbol{M}_b$ are mass matrices, $\boldsymbol{D}$ is the weak divergence term, and $\boldsymbol{Q}$ is an operator containing the vertical components of $\delta \boldsymbol{u}_h$. In the absence of orography,

---

[6]The choice for $W_h^b$ in (97) corresponds to a Charney-Phillips vertical staggering of the buoyancy variable, which is the desired approach for the UK Met Office's Unified Model (Melvin et al., 2010). One could also collocate $b_h$ with $p_h$ ($b_h \in W_h^3$), which corresponds to a Lorenz staggering. This however supports a computational mode which is exacerbated by fast-moving waves. We restrict our discussion to the former case.

we can use the point-wise expression for the buoyancy:

$$\delta b_h = r_b - \frac{\Delta t}{2} N^2 \delta \boldsymbol{u}_h \cdot \hat{\boldsymbol{z}} \tag{102}$$

and eliminate $\delta b_h$ from (101) by substituting (102) into (98). This produces the following mixed velocity-pressure system:

$$\mathcal{A} \begin{Bmatrix} U \\ P \end{Bmatrix} = \begin{bmatrix} \widetilde{\boldsymbol{A}}_{\boldsymbol{u}} & -\frac{\Delta t}{2} \boldsymbol{D}^T \\ c^2 \frac{\Delta t}{2} \boldsymbol{D} & \boldsymbol{M}_p \end{bmatrix} \begin{Bmatrix} U \\ P \end{Bmatrix} = \begin{Bmatrix} \widetilde{\boldsymbol{R}}_{\boldsymbol{u}} \\ \boldsymbol{R}_p \end{Bmatrix}, \tag{103}$$

where $\widetilde{\boldsymbol{A}}_{\boldsymbol{u}} = \boldsymbol{A}_{\boldsymbol{u}} + \frac{\Delta t^2}{4} N^2 \boldsymbol{Q}^T \boldsymbol{M}_b^{-1} \boldsymbol{Q}$ and $\widetilde{\boldsymbol{R}}_{\boldsymbol{u}} = \boldsymbol{R}_{\boldsymbol{u}} + \frac{\Delta t}{2} \boldsymbol{Q}^T \boldsymbol{M}_b^{-1} \boldsymbol{R}_b$ is the modified velocity operator and right-hand side respectively. Note that in our elimination strategy, $\widetilde{\boldsymbol{A}}_{\boldsymbol{u}}$ corresponds to the bilinear form obtained after eliminating the buoyancy at the equation level:

$$\widetilde{\boldsymbol{A}}_{\boldsymbol{u}} \leftarrow (\boldsymbol{w}, \delta \boldsymbol{u})_{\mathcal{T}_h} + \frac{\Delta t}{2} (\boldsymbol{w}, 2\boldsymbol{\Omega} \times \delta \boldsymbol{u})_{\mathcal{T}_h} + \frac{\Delta t^2}{4} N^2 (\boldsymbol{w} \cdot \hat{\boldsymbol{z}}, \delta \boldsymbol{u} \cdot \hat{\boldsymbol{z}})_{\mathcal{T}_h}. \tag{104}$$

A similar construction holds for $\widetilde{\boldsymbol{R}}_{\boldsymbol{u}}$. Once (103) is solved, $\delta b_h$ is reconstructed by solving:

$$\boldsymbol{M}_b \boldsymbol{B} = \boldsymbol{R}_b - \frac{\Delta t}{2} N^2 \boldsymbol{Q} \boldsymbol{U}. \tag{105}$$

Equation (105) can be efficiently inverted using the conjugate gradient method.

### 5.3.2 Preconditioning the mixed velocity pressure system

The primary difficulty is finding efficient solvers for (103). This was studied by Mitchell and Müller (2016) within the context of developing a preconditioner which is robust against parameters, like $\Delta t$ and mesh resolution. However, the implicit treatment of the Coriolis term was not taken into account. We consider two preconditioning strategies.

One strategy proposed by Mitchell and Müller (2016) is to build a preconditioner based on the Schur-complement factorization of $\mathcal{A}$ in (103):

$$\mathcal{A}^{-1} = \begin{bmatrix} \boldsymbol{I} & \frac{\Delta t}{2} \widetilde{\boldsymbol{A}}_{\boldsymbol{u}}^{-1} \boldsymbol{D}^T \\ \boldsymbol{0} & \boldsymbol{I} \end{bmatrix} \begin{bmatrix} \widetilde{\boldsymbol{A}}_{\boldsymbol{u}}^{-1} & \boldsymbol{0} \\ \boldsymbol{0} & \boldsymbol{H}^{-1} \end{bmatrix} \begin{bmatrix} \boldsymbol{I} & \boldsymbol{0} \\ -c^2 \frac{\Delta t}{2} \boldsymbol{D} \widetilde{\boldsymbol{A}}_{\boldsymbol{u}}^{-1} & \boldsymbol{I} \end{bmatrix}, \tag{106}$$

where $\boldsymbol{H} = \boldsymbol{M}_p + c^2 \frac{\Delta t^2}{4} \boldsymbol{D} \widetilde{\boldsymbol{A}}_{\boldsymbol{u}}^{-1} \boldsymbol{D}^T$ is the dense pressure Helmholtz operator. Because we have chosen to include the Coriolis term, the operator $\boldsymbol{H}$ is non-symmetric and has the form:

$$\boldsymbol{H} = \boldsymbol{M}_p + c^2 \frac{\Delta t^2}{4} \boldsymbol{D} \left( \widetilde{\boldsymbol{M}}_{\boldsymbol{u}} + \frac{\Delta t}{2} \boldsymbol{C}_{\boldsymbol{\Omega}} \right)^{-1} \boldsymbol{D}^T, \tag{107}$$

where $\widetilde{\boldsymbol{M}}_{\boldsymbol{u}}$ is a modified mass matrix. As $\Delta t$ increases, the contribution of $\boldsymbol{C}_{\boldsymbol{\Omega}}$ becomes more prominent in $\boldsymbol{H}$, making sparse approximations of $\boldsymbol{H}$ more challenging. We shall elaborate on this further below when we present the results of our second solver strategy.

Our preferred strategy solves the hybridizable formulation of the system eqrefeq:mixed-vel-pr-system. Let $W_h^{2,d}$ denote the broken version of $W_h^2$ and $M_h$ the space of Lagrange multipliers. Then the hybridizable formulation for the velocity-pressure system reads as follows: find $\delta \boldsymbol{u}_h^d \in W_h^{2,d}$, $\delta p_h \in W_h^3$, and $\lambda_h \in M_h$ such that

$$
\left(\boldsymbol{w}, \delta \boldsymbol{u}_h^d\right)_{\mathcal{T}_h} + \frac{\Delta t}{2}\left(\boldsymbol{w}, 2\boldsymbol{\Omega} \times \delta \boldsymbol{u}_h^d\right)_{\mathcal{T}_h} + \frac{\Delta t^2}{4}N^2\left(\boldsymbol{w}\cdot\hat{\boldsymbol{z}}, \delta \boldsymbol{u}_h^d \cdot \hat{\boldsymbol{z}}\right)_{\mathcal{T}_h}
$$

$$
-\frac{\Delta t}{2}\left(\nabla \cdot \boldsymbol{w}, \delta p_h\right)_{\mathcal{T}_h} + \left\langle [[\boldsymbol{w}]], \lambda_h \right\rangle_{\partial \mathcal{T}_h} = \widetilde{r}_u, \quad \forall \boldsymbol{w} \in W_h^{2,d} \tag{108}
$$

$$
\left(\phi, \delta p_h\right)_{\mathcal{T}_h} + \frac{\Delta t}{2}c^2\left(\phi, \nabla \cdot \delta \boldsymbol{u}_h\right)_{\mathcal{T}_h} = r_p, \quad \forall \phi \in W_h^3, \tag{109}
$$

$$
\left\langle \gamma, \left[\left[\delta \boldsymbol{u}_h^d\right]\right]\right\rangle_{\partial \mathcal{T}_h} = 0, \quad \forall \gamma \in M_h, \tag{110}
$$

The system (108)–(110) is automatically formed by the Firedrake preconditioner: `firedrake.HybridizationPC`. We then locally eliminate the velocity and pressure after hybridization, producing the condensed problem:

$$
\boldsymbol{H}_\partial \boldsymbol{\Lambda} = \boldsymbol{E}, \quad \boldsymbol{H}_\partial = \begin{bmatrix} \boldsymbol{C} & \boldsymbol{0} \end{bmatrix} \begin{bmatrix} \widehat{\widehat{\boldsymbol{A}}}_{\boldsymbol{u}} & -\frac{\Delta t}{2}\widehat{\boldsymbol{D}}^T \\ c^2\frac{\Delta t}{2}\widehat{\boldsymbol{D}} & \boldsymbol{M}_p \end{bmatrix}^{-1} \begin{bmatrix} \boldsymbol{C}^T \\ \boldsymbol{0} \end{bmatrix}, \quad E = \begin{bmatrix} \boldsymbol{C} & \boldsymbol{0} \end{bmatrix} \begin{bmatrix} \widehat{\widehat{\boldsymbol{A}}}_{\boldsymbol{u}} & -\frac{\Delta t}{2}\widehat{\boldsymbol{D}}^T \\ c^2\frac{\Delta t}{2}\widehat{\boldsymbol{D}} & \boldsymbol{M}_p \end{bmatrix}^{-1} \left\{ \begin{matrix} \widehat{R}_{\boldsymbol{u}} \\ R_p \end{matrix} \right\}, \tag{111}
$$

where $\widehat{\cdot}$ denotes matrices/vectors with discontinuous test and trial functions. The non-symmetric operator $\boldsymbol{H}_\partial$ is inverted using a preconditioned generalized conjugate residual (GCR) Krylov method, as suggested in Thomas et al. (2003). For our choice of preconditioner, we follow strategies outlined in Elman et al. (2001) and employ an algebraic multigrid method (V-cycle) with GMRES (five iterations) smoothers on the coarse levels. The GMRES smoothers are preconditioned with block ILU on
each level. For the finest level, block ILU produces a line smoother (necessary for efficient solution on thin domains) when the trace variable nodes are numbered in vertical lines, as is the case in our Firedrake implementation. On the coarser levels, less is known about the properties of ILU under the AMG coarsening strategies, but as we shall see, we observe performance that suggests ILU is still behaving as a line smoother. More discussion on multigrid for non-symmetric problems can be found in Bramble et al. (1994, 1988); Mandel (1986). A gravity wave test using our solution strategy and hybridization preconditioner
is illustrated in Figure 7 for a problem on a condensed Earth (radius scaled down by a factor of 125) and 10km lid.

### 5.3.3 Robustness against acoustic Courant number with implicit Coriolis

In this final experiment, we repeat a similar study to that presented in Mitchell and Müller (2016). Our setup closely resembles the gravity wave test of Skamarock and Klemp (1994) extended to a spherical annulus. We initialize the velocity in a simple solid-body rotation and introduce a localized buoyancy anomaly. A Coriolis term is introduced as a function of the Cartesian coordinate $z$, and is constant along lines of latitude ($f$-plane): $2\boldsymbol{\Omega} = 2\Omega_r \frac{z}{R}\hat{\boldsymbol{z}}$, with angular rotation rate $\Omega_r = 7.292 \times 10^{-5}\mathrm{s}^{-1}$.
We fix the resolution of the problem and run the solver over a range of $\Delta t$. We measure this by adjusting the horizontal acoustic Courant number $\lambda_C = c\frac{\Delta t}{\Delta x}$, where $c$ is the speed of sound and $\Delta x$ is the horizontal resolution.

Note that the range of Courant numbers used in this paper exceeds what is typical in operational forecast settings (typically between $\mathcal{O}(2)$–$\mathcal{O}(10)$). The grid set up mirrors that of actual global weather models; we extrude a spherical mesh of the Earth

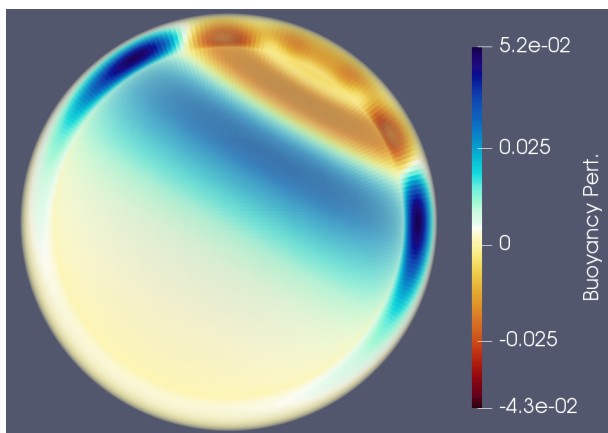

**Figure 7.** Buoyancy perturbation ($y - z$ cross section) at $t = 3600$s from a simple gravity wave test ($\Delta t = 100$s). The initial conditions (in lat.-long. coordinates) for the velocity is a simple solid-body rotation: $\boldsymbol{u} = 20\boldsymbol{e}_\lambda$, where $\boldsymbol{e}_\lambda$ is the unit vector pointing in the direction of decreasing longitude. A buoyancy anomaly is defined via $b = \frac{d^2}{d^2+q^2} \sin(\pi z/10000)$, where $q = R\cos^{-1}(\cos(\phi)\cos(\lambda - \lambda_\phi))$, $d = 5000$m, $R = 6371$km$/125$ is the planet radius, and $\lambda_\phi = 2/3$. The equations are discretized using the lowest-order method RTCF$_1$, with 24,576 quadrilateral cells in the horizontal and 64 extrusion levels. The velocity-pressure system is solved using hybridization.

**Table 6.** Grid set up and discretizations for the acoustic Courant number study. The total unknowns (velocity and pressure) and hybridized unknowns (broken velocity, pressure, and trace) are shown in the last two columns (millions). The vertical resolution is fixed across all discretizations.

| **Discretizations and grid information** | | | | | | |
|:---:|:---:|:---:|:---:|:---:|:---:|:---:|
| Mixed method | # horiz. cells | # vert. layers | $\Delta x$ | $\Delta z$ | $U$-$P$ dofs | Hybrid. dofs |
| RT$_1$ | 81,920 | 85 | 86 km | 1,000 m | 24.5 M | 59.3 M |
| RT$_2$ | 5,120 | 85 | 346 km | 1,000 m | 9.6 M | 17.4 M |
| BDFM$_2$ | 5,120 | 85 | 346 km | 1,000 m | 10.5 M | 18.3 M |
| RTCF$_1$ | 98,304 | 85 | 78 km | 1,000 m | 33.5 M | 83.7 M |
| RTCF$_2$ | 6,144 | 85 | 312 km | 1,000 m | 16.7 M | 29.3 M |

upwards to a height of 85km. The set up for the different discretizations (including degrees of freedom for the velocity-pressure and hybridized systems) are presented in Table 6.

It was shown in Mitchell and Müller (2016) that using a sparse approximation of the pressure Schur-complement of the form:

$$\widetilde{\boldsymbol{H}} = \boldsymbol{M}_p + c^2 \frac{\Delta t^2}{4} \boldsymbol{D} \left( \mathrm{Diag}(\widetilde{\boldsymbol{A}}_{\boldsymbol{u}}) \right)^{-1} \boldsymbol{D}^T \tag{112}$$

served as a good preconditioner, leading to a system that was amenable to multigrid methods and resulted in a Courant number independent solver. However, when the Coriolis term is included, this is no longer the case: the diagonal approximation $\mathrm{Diag}(\widetilde{\boldsymbol{A}}_{\boldsymbol{u}})$ becomes worse with increasing $\lambda_C$. To demonstrate this, we solve the gravity wave system on a low-resolution grid

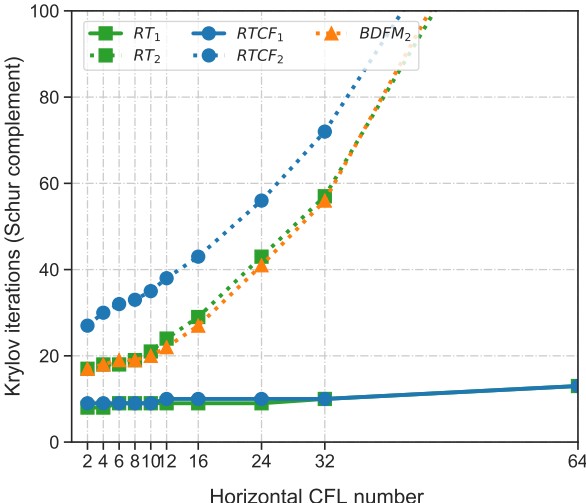

**Figure 8.** Number of Krylov iterations to invert the Helmholtz system using $\widetilde{H}^{-1}$ as a preconditioner. The preconditioner is applied using a direct LU factorization within a GMRES method on the entire pressure equation. While the lowest-order methods grow slowly over the Courant number range, the higher-order (by only one approximation order) methods quickly degrade and diverge after the critical range $\lambda_C = \mathcal{O}(2)–\mathcal{O}(10)$. At $\lambda_C > 32$, the solvers take over 150 iterations.

(10km lid, 10 vertical levels, maintaining the same cell aspect ratio as in Table 6) using the Schur-complement factorization (106). LU factorizations are applied to invert both $\widetilde{A}_u^{-1}$ and $\widetilde{H}^{-1}$. Inverting the Schur-complement $H^{-1}$ is done using precon-ditioned GMRES iterations, and a flexible-GMRES algorithm is used on the full velocity-pressure system. If $\widetilde{H}^{-1}$ is a good approximation to $H^{-1}$, then we should see low iteration counts in the Schur-complement solve. Figure 8 shows the results of
this study for a range of Courant numbers.

For the lower-order methods, the number of iterations to invert $H$ grow slowly but remain under control. Increasing the approximation degree by one results in degraded performance. As $\Delta t$ increases, the number of Krylov iterations needed to invert the system to a relative tolerance of $10^{-5}$ grows rapidly. It is clear that this sparse approximation is not robust against Courant number. This can be explained by the fact that diagonalizing the velocity operator fails to take into account the effects
of the Coriolis term (which appear in off-diagonal positions in the operator). Even if one were to use traditional mass-lumping (row-wise sums), the Coriolis effects are effectively cancelled out due to asymmetry.

Hybridization avoids this problem entirely: we always construct an exact Schur complement, and only have to worry about solving the trace system (111). We now show that this approach (described in Section 5.3.2) is much more robust to changes in $\Delta t$. We use the same workstation as for the three-dimensional CG/HDG problem in Section 5.1 (executed with a total of 40
MPI processes). Figure 9 shows the parameter test for all the discretizations described in Table 6. We see that, in terms of total number of GCR iterations needed to invert the trace system, hybridization is far more controlled as Courant number increases. They largely remain constant throughout the entire parameter range, only varying by an iteration or two. It is not until after

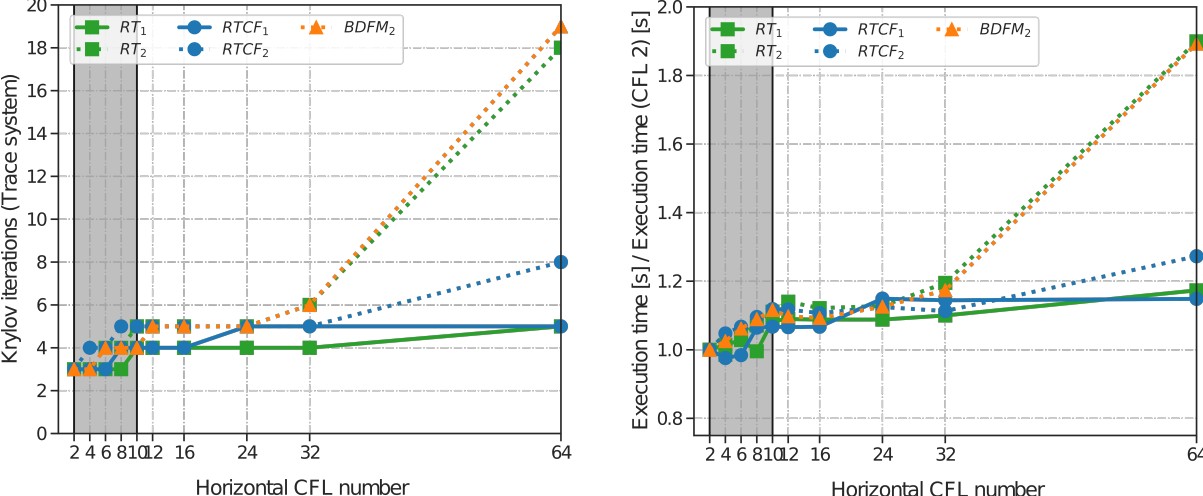

**Figure 9.** Courant number parameter test run on a fully-loaded compute node. Both figures display the hybridized solver for each discretization, described in Table 6. The left figure displays to total iteration count (preconditioned GCR) to solve the trace system to a relative tolerance of $10^{-5}$. The right figure displays the relative work of each solver, which takes into account the time required to forward eliminate and locally recover the velocity and pressure.

$\lambda_C > 32$ that we begin to see a larger jump in the number of GCR iterations. This is expected, since the Coriolis operator causes the problem to become singular for very large Courant numbers. However, unlike with the approximate Schur-complement solver, iteration counts are still under control. In particular, each method (lowest and higher order) remains constant throughout the critical range (shaded in gray in Figure 9).

In Figure 9 (right), we display the ratio of execution time and the time-to-solution at the lowest Courant number of two. We perform this normalization to better compare the lower and higher order methods (and discretizations on triangular prisms vs extruded quadrilaterals). The calculation of the ratios includes the time needed to eliminate and reconstruct the hybridized velocity/pressure variables. The fact that the hybridization solver remains close to one demonstrates that the entire solution procedure is largely $\lambda_C$-independent until around $\lambda_C = 32$. The overall trend is largely the same as the number of Krylov

iterations to reach solver convergence. This is due to our hybridization approach being solver dominated, with local operations like forward elimination together with local recovery taking approximately $1/3$ of the total execution time for each method. The percentage breakdown of the hybridization solver is similar to what is already presented in Section 5.1.2.

    Implicitly treating the Coriolis term has been discussed for semi-implicit discretizations of large-scale geophysical flows (Temperton, 1997; Cullen, 2001; Nechaev and Yaremchuk, 2004). The addition of the Coriolis term presents a particular

challenge to the solution finite element discretizations of these equations since it increases the difficulty of finding a good sparse approximation of the nonsymmetric elliptic operator. Hybridization shows promise here, as it allows for the assembly of the elliptic equation that both captures the effects of rotation and results in a sparse linear system.

# 6 Conclusions

We have presented Slate, and shown how this language can be used to create concise mathematical representations of localized linear algebra on the tensors corresponding to finite element forms. We have shown how this DSL can be used in tandem with UFL in Firedrake to implement solution approaches making use of automated code generation for static condensation, hybridization, and localized post-processing. Setup and configuration is done at runtime, allowing one to switch in different discretizations at will. In particular, this framework alleviates much of the difficulty in implementing such methods within intricate numerical, and paves the way for future low-level optimizations. In this way, the framework in this paper can be used to help enable the rapid development and exploration of new hybridization and static condensation techniques for a wide class of problems. We remark here that the reduction of global matrices via element-wise algebraic static condensation, as described in Guyan (1965); Irons (1965) is also possible using Slate, including other more general static condensation procedures outside the context of hybridization.

Our approach to preconditioner design revolves around its composable nature, in that these Slate-based implementations can be seamlessly incorporated into complicated solution schemes. In particular, there is current research in the design of dynamical cores for numerical weather prediction using implementations of hybridization and static condensation with Slate (Bauer and Cotter, 2018; Shipton et al., 2018). The performance of such methods for geophysical flows are a subject of ongoing investigation.

In this paper, we have provided some examples of hybridization procedures for compatible finite element discretizations of geophysical flows. These approaches avoid the difficulty in constructing sparse approximations of dense elliptic operators. Static condensation arising from hybridizable formulations can best be interpreted as producing an *exact* Schur-complement factorization on the global hybridizable system. This eliminates the need for outer iterations from a suitable Krylov method to solve the full mixed system, and replaces the original global mixed equations with a condensed elliptic system. More extensive performance benchmarks, which requires detailed analysis of the resulting operator systems arising from hybridization, is a necessary next-step to determine whether hybridization provides a scalable solution strategy for compatible finite elements in operational settings.

*Code and data availability.* The contribution in this paper is available through open-source software provided by the Firedrake Project: https://www.firedrakeproject.org/. We cite archives of the exact software versions used to produce the results in this paper. For all components of the Firedrake project used in this paper, see Zenodo/Firedrake (2019). The numerical experiments, full solver configurations, code-verification (including local-processing), and raw data are available in Zenodo/Tabula-Rasa (2019).

## Appendix A: Semi-implicit method for the shallow water system

For some tessellation, $\mathcal{T}_h$, our semi-discrete mixed method for (81)–(82) seeks approximations $(\boldsymbol{u}_h, D_h) \in U_h \times V_h \subset H(\mathrm{div}) \times L^2$ satisfying:

$$
\left(\boldsymbol{w}, \frac{\partial \boldsymbol{u}_h}{\partial t}\right)_{\mathcal{T}_h} - \left(\nabla^\perp\left(\boldsymbol{w} \cdot \boldsymbol{u}_h^\perp\right), \boldsymbol{u}_h^\perp\right)_{\mathcal{T}_h} + \left(\boldsymbol{w}, f\boldsymbol{u}_h^\perp\right)_{\mathcal{T}_h} + \left\langle [[\boldsymbol{n}^\perp \boldsymbol{w} \cdot \boldsymbol{u}_h^\perp]], \tilde{\boldsymbol{u}}_h^\perp \right\rangle_{\partial \mathcal{T}_h}
$$
$$
- \left(\nabla \cdot \boldsymbol{w}, g\left(D_h + b\right) + \frac{1}{2}|\boldsymbol{u}_h|^2\right)_{\mathcal{T}_h} = 0, \quad \forall \boldsymbol{w} \in U_h, \tag{A1}
$$

$$
\left(\phi, \frac{\partial D_h}{\partial t}\right)_{\mathcal{T}_h} - \left(\nabla\phi, \boldsymbol{u}_h D_h\right)_{\mathcal{T}_h} + \left\langle [[\phi \boldsymbol{u}_h]], \tilde{D}_h \right\rangle_{\partial \mathcal{T}_h}, = 0, \quad \forall \phi \in V_h, \tag{A2}
$$

where $\tilde{\phantom{i}}$ indicates that the value of the function should be taken from the upwind side of each facet. The discretisation of the velocity advection operator is an extension of the energy-conserving scheme of Natale and Cotter (2017) to the shallow-water equations.

The time-stepping scheme follows a Picard iteration semi-implicit approach, where predictive values of the relevant fields are determined via an explicit step of the advection equations, and corrective updates are generated by solving an implicit linear system (linearized about a state of rest) for $(\Delta \boldsymbol{u}_h, \Delta D_h) \in \boldsymbol{U}_h \times V_h$, given by

$$
(\boldsymbol{w}, \Delta \boldsymbol{u}_h)_{\mathcal{T}_h} + \frac{\Delta t}{2}\left(\boldsymbol{w}, f\Delta \boldsymbol{u}_h^\perp\right)_{\mathcal{T}_h} - \frac{\Delta t}{2}\left(\nabla \cdot \boldsymbol{w}, g\Delta D_h\right)_{\mathcal{T}_h} = -R_{\boldsymbol{u}}[\boldsymbol{u}_h^{n+1}, D_h^{n+1}; \boldsymbol{w}], \quad \forall \boldsymbol{w} \in U_h, \tag{A3}
$$

$$
(\phi, \Delta D_h)_{\mathcal{T}_h} + \frac{H\Delta t}{2}\left(\phi, \nabla \cdot \Delta \boldsymbol{u}_h\right)_{\mathcal{T}_h} = -R_D[\boldsymbol{u}_h^{n+1}, D_h^{n+1}; \phi], \quad \forall \phi \in V_h, \tag{A4}
$$

where $H$ is the mean layer depth, and $R_{\boldsymbol{u}}$ and $R_D$ are residual linear forms that vanish when $\boldsymbol{u}_h^{n+1}$ and $D_h^{n+1}$ are solutions to the implicit midpoint rule time discretization of (A1)–(A2). The residuals are evaluated using the predictive values of $\boldsymbol{u}_h^{n+1}$ and $D_h^{n+1}$.

The implicit midpoint rule time discretization of the nonlinear rotating shallow water equations (A1)–(A2) is:

$$
\left(\boldsymbol{w}, \boldsymbol{u}_h^{n+1} - \boldsymbol{u}_h^n\right)_{\mathcal{T}_h} - \Delta t\left(\nabla^\perp\left(\boldsymbol{w} \cdot \boldsymbol{u}_h^{*\perp}\right), \boldsymbol{u}_h^{*\perp}\right)_{\mathcal{T}_h} + \Delta t\left(\boldsymbol{w}, f\boldsymbol{u}_h^{*\perp}\right)_{\mathcal{T}_h}
$$
$$
+ \Delta t\left\langle [[\boldsymbol{n}^\perp \boldsymbol{w} \cdot \boldsymbol{u}_h^{*\perp}]], \tilde{\boldsymbol{u}}_h^{*\perp} \right\rangle_{\partial \mathcal{T}_h}
$$
$$
- \Delta t\left(\nabla \cdot \boldsymbol{w}, g\left(D_h^* + b\right) + \frac{1}{2}|\boldsymbol{u}_h^*|^2\right)_{\mathcal{T}_h} = 0, \quad \forall \boldsymbol{w} \in U_h, \tag{A5}
$$

$$
\left(\phi, D_h^{n+1} - D_h^n\right)_{\mathcal{T}_h} - \Delta t\left(\nabla\phi, \boldsymbol{u}_h^* D_h^*\right)_{\mathcal{T}_h} + \Delta t\left\langle [[\phi \boldsymbol{u}_h^*]], \tilde{D}_h^* \right\rangle_{\partial \mathcal{T}_h}, = 0, \quad \forall \phi \in V_h, \tag{A6}
$$

where $\boldsymbol{u}_h^* = (\boldsymbol{u}_h^{n+1} + \boldsymbol{u}_h^n)/2$ and $D_h^* = (D_h^{n+1} + D_h^n)/2$.

One approach to construct the residual functionals $R_{\boldsymbol{u}}$ and $R_D$ would be to simply define these from (A5)–(A6). However, this leads to a small critical time-step for stability of the scheme. To make the numerical scheme more stable, we define

residuals as follows. For $R_{\boldsymbol{u}}$, we first solve for $\boldsymbol{v}_h \in \boldsymbol{U}_h$ such that

$$(\boldsymbol{w}, \boldsymbol{v}_h - \boldsymbol{u}_h^n)_{\mathcal{T}_h} - \Delta t \left(\nabla^\perp \left(\boldsymbol{w} \cdot \boldsymbol{u}_h^{*\perp}\right), \boldsymbol{v}_h^{\sharp\perp}\right)_{\mathcal{T}_h} + \Delta t \left(\boldsymbol{w}, f\boldsymbol{v}_h^{\sharp\perp}\right)_{\mathcal{T}_h}$$
$$+ \Delta t \left\langle \left[\!\left[\boldsymbol{n}^\perp \boldsymbol{w} \cdot \boldsymbol{u}_h^{*\perp}\right]\!\right], \tilde{\boldsymbol{v}}_h^{\sharp\perp}\right\rangle_{\partial\mathcal{T}_h}$$
$$- \Delta t \left(\nabla \cdot \boldsymbol{w}, g\left(D_h^* + b\right) + \frac{1}{2}|\boldsymbol{u}_h^*|^2\right)_{\mathcal{T}_h} = 0, \quad \forall \boldsymbol{w} \in U_h, \tag{A7}$$

where $\boldsymbol{v}_h^\sharp = (\boldsymbol{v}_h + \boldsymbol{u}_h^n)/2$. This is a linear variational problem. Then,

$$R_{\boldsymbol{u}}[\boldsymbol{u}_h^{n+1}, D_h^{n+1}; \boldsymbol{w}] = \left(\boldsymbol{w}, \boldsymbol{v}_h - \boldsymbol{u}_h^{n+1}\right)_{\mathcal{T}_h}. \tag{A8}$$

Similarly, for $R_D$ we first solve for $E_h \in V_h$ such that

$$(\phi, E_h - D_h^n)_{\mathcal{T}_h} - \Delta t \left(\nabla\phi, \boldsymbol{u}_h^* E_h^\sharp\right)_{\mathcal{T}_h} + \Delta t \left\langle [\![\phi \boldsymbol{u}_h^*]\!], \tilde{E}_h^\sharp\right\rangle_{\partial\mathcal{T}_h} = 0, \quad \forall \phi \in V_h, \tag{A9}$$

where $E_h^\sharp = (E_h + D_h^n)/2$. This is also a linear problem. Then,

$$R_D[\boldsymbol{u}_h^{n+1}, D_h^{n+1}; \phi] = \left(\phi, E_h - D_h^{n+1}\right)_{\mathcal{T}_h}. \tag{A10}$$

    This process can be thought of as iteratively solving for the average velocity and depth that satisfies the implicit midpoint rule discretisation. Both (A7) and (A9) can be solved separately, since there is no coupling between them. The fields $\boldsymbol{v}_h$ and $E_h$ are then used to construct the right-hand side for the implicit linearized system in (A3)–(A4). Once the system is solved, the solution $(\Delta \boldsymbol{u}_h, \Delta D_h)$ is then used to update the iterative values of $\boldsymbol{u}_h^{n+1}$ and $D_h^{n+1}$ according to $(\boldsymbol{u}_h^{n+1}, D_h^{n+1}) \leftarrow$
$(\boldsymbol{u}_h^{n+1} + \Delta\boldsymbol{u}_h, D_h^{n+1} + \Delta D_h)$, having initially chosen $(\boldsymbol{u}_h^{n+1}, D_h^{n+1}) = (\boldsymbol{u}_h^n, D_h^n)$.

*Author contributions.* T. H. Gibson is the principal author and developer of the software presented in this paper and main author of the text. Authors L. Mitchell and D. A. Ham assisted and guided the software abstraction as a domain-specific language, and edited text. C. J. Cotter contributed to the formulation of the geophysical fluid dynamics and the design of the numerical experiments, and edited text.

*Competing interests.* D. A. Ham is an executive editor of the journal. The other authors declare they have no other competing interests.

*Acknowledgements.* This work was supported by the Engineering and Physical Sciences Research Council (grant numbers: EP/M011054/1, EP/L000407/1, and EP/L016613/1), and the Natural Environment Research Council (grant number: NE/K008951/1).

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
