# Peer review of "Slate: extending Firedrake's domain-specific abstraction to hybridized solvers for geoscience and beyond"

_Geoscientific Model Development, 2019_

## Short Comment (SC1) · 17 May 2019

Dear authors,

in my role as Executive editor of GMD, I would like to bring to your attention our Editorial version 1.1:

http://www.geosci-model-dev.net/8/3487/2015/gmd-8-3487-2015.html

This highlights some requirements of papers published in GMD, which is also available on the GMD website in the 'Manuscript Types' section:

http://www.geoscientific-model-development.net/submission/manuscript_types.html

In particular, please note that for your paper, the following requirements have not been met in the Discussions paper:

- "The main paper must give the model name and version number (or other unique identifier) in the title."

In order to simplify reference to your developments, please add a version number for SLATE in the title of your article in your revised submission to GMD.

Yours,

Astrid Kerkweg

---

## Author Comment (AC1) · 20 May 2019

Dear Executive Editor,

Thank you for your comment. We are aware that GMD requires a version number in the manuscript title. However, this software is part of the Firedrake package and does not use version numbers. This is due to that fact that Firedrake operates on a continuous release system. This implies that the only supported version of Firedrake and all of its components (including Slate) is the one available on the master branch on Github.

A similar issue was raised and resolved in another paper within the Firedrake special

issue. The discussion can be found here: https://www.geosci-model-dev.net/11/4359/2018/gmd-11-4359-2018-discussion.html

While Firedrake does not use version numbers, we do provide exact software versions used to produce the results in the paper (labeled by a corresponding Zenodo DOI). This ensures reproducibility as per the GMD standard. We also provide a facility for Firedrake users to generate DOIs for the exact set of Firedrake components used in an experiment. This identifies the version of the software used much more accurately than a version number would.

We would like to request that the title remains as is.

On behalf of the authors, Thomas Gibson

---

## Referee Comment (RC1) · Anonymous Referee #1 · 29 May 2019

This is a strong contribution that combines state-of-the-art (albeit existing) numerical approaches of great interest to the geosciences community, with a well-engineered and extendable software implementation (Slate).

The authors state that the contribution is in the automated translation of mathematics to compiled code, and on this the paper delivers. The topic is clearly suitable within the context of the Firedrake special issue. Having implemented cell local operations manually using Eigen and C++, I can attest to the usefulness of having something like Slate available to the community.

It would be useful to have the following questions addressed in the manuscript:

[Figure]

1) How is the .inv() command translated to generated code? Are you using LU/Cholesky or directly inverting the matrix? It is clear the former for the A.solve() command as the decomposition is passed, but not for .inv(). 2) How would you deal in practice with non-linear problems solved using, e.g. a Newton-Krylov method. Is it possible to compose SLEPc and the Preconditioned Krylov Solvers? How would one setup SLEPc to call the code to recover the internal/local variables between Newton iterations? 3) What does the generated Eigen code look like? 4) It is not totally clear how the output from TSFC is fed into the linear algebra compiler. Do you call the TSFC kernel, get the complete cell tensor, split it and then perform the dense linear algebra operations? Or is everything 'interleaved' into one single cell tensor kernel by the linear algebra compiler? Or do you call multiple TSFC kernels, one for each sub-block and then perform the dense linear algebra operations? 5) The high-level problem setup in sections 1 and 2 is pretty terse. I don't think someone who has some knowledge of FEM but is not a real subject expert could get through this section. Some of the wording is quite heavy on jargon too, e.g global data structure $\sim$ sparse matrix? I appreciate you don't have time to do a deep-dive into FE, but a bit more text and some pointers to more detailed explanations (other Firedrake papers?) would be useful.

Small notational comments:

* You do not mention the bold symbol = vector function convention. * Your convention of non-bold capitals being (local? - not sure) discrete linear operators (matrices and vectors) is also not mentioned, although K (the finite element cell) would break this 'convention'. I'd also note that K is used twice for different objects (the finite element cell and a matrix in eq.76) It would probably aid readability if it was possible to distinguish between matrices and vectors, global and local, especially when the operations become more complex in the later sections. There also seem to be (global? - not sure) discrete linear operators in bold.

---

## Referee Comment (RC2) · Anonymous Referee #2 · 7 Oct 2019

**1. GENERAL COMMENTS**

This is an excellent, well-written paper describing an important contribution to finite element software. Among numerical analysts and practitioners, there has been increasing interest in hybridization, static condensation, and post-processing of mixed and (more recently) discontinuous Galerkin finite element methods. However, these methods have been difficult to implement for users of standard finite element software. The authors have developed and documented a high-level domain specific language, called Slate, along with an implementation that makes these techniques widely accessible within the Firedrake finite element package. This is comparable to how the Unified

Form Language (UFL), underlying both Firedrake and FEniCS, made it possible for finite element variational problems to be described at a high level. The authors include numerical examples illustrating the application of Slate, and the numerical results agree with theory.

2. SPECIFIC COMMENTS

There are only a few minor revisions that I would suggest.

2.1. There are a few inaccuracies and omissions in the historical references for hybridization of mixed finite element methods, particularly in the introduction.

2.1.1. The bibliographic reference to Brezzi and Fortin's book (from 1991, not 2012) is incorrect. The correct reference is:

F. Brezzi and M. Fortin, Mixed and hybrid finite element methods, vol. 15 of Springer Series in Computational Mathematics, Springer-Verlag, New York, 1991.

The authors may also wish to cite the following, which is essentially an updated version of the Brezzi and Fortin book:

D. Boffi, F. Brezzi, and M. Fortin, Mixed finite element methods and applications, vol. 44 of Springer Series in Computational Mathematics, Springer, Heidelberg, 2013.

2.1.2. Hybridization and static condensation of mixed methods was apparently first introduced in the following reference:

B. M. Fraejis de Veubeke, Displacement and equilibrium models in the finite element method, in Stress Analysis, O. Zienkiewicz and G. Holister, eds., Wiley, New York, 1965. Reprinted in Internat. J. Numer. Methods Engrg., 52 (2001), pp. 287–342.

2.1.3. Local post-processing appeared in Arnold and Brezzi (1985) for the hybridized RT method and in Brezzi, Douglas, and Marini (1985) for the hybridized BDM method. The authors frequently cite the former but not the latter when mentioning hybridization and post-processing of mixed methods. Also, the 1991 paper of Stenberg on postprocessing is cited elsewhere in the paper but not in the introduction. I believe that these three papers should be cited in the relevant part of the introduction (p. 2, l. 18).

2.2. Section 2.1 is somewhat confusing and could be improved.

2.2.1. The notation $a(\mathbf{c}; \mathbf{v})$ originally made me think that $a(\cdot; \cdot)$ was a bilinear form. It took a few times through before I made sense of the notation and realized how linear, bilinear, etc., forms are specified. It might be helpful to include one or two concrete examples before introducing a "general form." This is done nicely in Section 2.1 of Alnaes et al. (2014), which I suggest that the authors emulate.

2.2.2. The notation for the $\mathcal{I}^c$ and $\mathcal{I}^f$ integrals is also confusing. Presumably $c$ stands for "cell" and $f$ for "facet," but at first I thought $c$ was the coefficient function $\mathbf{c}$ and $f$ was some source function, as in equation (10). Since the authors have been using $\mathcal{T}_h$ for cells and $\mathcal{E}_h$ for facets, perhaps a clearer notation would be to call these $\mathcal{I}^{\mathcal{T}}$, $\mathcal{I}^{\mathcal{E},\circ}$, and $\mathcal{I}^{\mathcal{E},\partial}$. A short sentence mentioning that the three $\mathcal{I}$s correspond to the contributions from the cells, internal facets, and boundary facets, respectively, would make this easier for the reader to understand.

3. TECHNICAL CORRECTIONS

I have no technical corrections to suggest, other than to correct the bibliographic reference to the book by Brezzi and Fortin, as described above.

—————————————————

---

## Author Comment (AC2) · 28 Oct 2019

The authors would like to sincerely thank the referee for the careful review and constructive comments for improving the manuscript.

1) "How is the `.inv()` command translated to generated code? Are you using LU/Cholesky or directly inverting the matrix? It is clear the former for the `A.solve()` command as the decomposition is passed, but not for `.inv()`."

This is a good question, and we will make this more clear in the updated version of the paper. As mentioned on page 6, Slate expressions are transformed into C++ code

using Eigen as the main linear algebra interface. By default, `A.inv()` gets translated to the corresponding Eigen call: `.inverse()`, which uses LU with partial pivoting for general matrices. It is also the default behavior of `A.solve(B)`. A complete list of factorization strategies compatible with the current version of Slate are documented on Eigen's public webpage (`https://eigen.tuxfamily.org/dox/`), under the category: "Dense linear problems and decompositions." An example of how one would specify a factorization type directly is shown in Listing 1, lines 28 and 30.

2) "How would you deal in practice with non-linear problems solved using, e.g., a Newton-Krylov method? Is it possible to compose SLEPc and the Preconditioned Krylov Solvers? How would one setup SLEPc to call the code to recover the internal/local variables between Newton iterations?"

If you mean "can we differentiate through Slate expressions?" for problems with local non-linear systems, this is not currently supported. However, one can imagine extending the Slate abstraction to support non-linear solvers. The question then becomes: "how do we generate code for this?" This is a topic of on-going discussion within the Firedrake project, and therefore, something like this may very well become part of the core Slate abstraction.

For global non-linear problems, we don't need to do anything extra in Slate. One can use the technology provided in Section 4 of the paper together with PETSc/SLEPc non-linear solvers. Standard Newton, for example, requires the solution of the linearized Jacobian system for the linear updates. Each Newton iteration, therefore, requires a linear solve. In Firedrake, the Jacobian is constructed by differentiating the non-linear PDE (expressed in UFL), which generates yet another UFL expression for the Jacobian. The result is then used as the operator for the linear solver (in PETSc, these are "KSP"s). Hybridization or static condensation can then be used by specifying the correct solver options. In practice, a single Newton iteration would look very similar to the Picard method described in Section 5.2. By design, all preconditioners presented in Section 4 are entirely composable with the PETSc
library and can be applied in the usual way, even when nested inside of an outer non-linear method (e.g. `snes_type newtonls`, `ksp_type preonly`, `pc_type python`, `pc_python_type firedrake.SCPC`). This follows from the framework developed in (Kirby & Mitchell, 2018). We shall stress this more clearly in the revised manuscript.

3) "What does the generated Eigen code look like?"

A .zip file is attached as a supplement (`static_condensation.zip`), which provides an entire generated C++ code and a PDF displaying just the kernel for static condensation (all generated from Slate). This was generated by an example taken directly from the shallow water test case in Section 5.2.

Slate's linear algebra compiler generates templated C++ code conforming to PyOP2's application program interface, described in (Rathgeber et al., 2012). Typically, the generated kernel will take an output tensor, any coefficients, and possibly external data as arguments. The body of the kernel will make appropriate function calls to local assembly kernels generated by Firedrake's form compiler TSFC (Homolya et al., 2018). Local data structures (element matrices/vectors) are derived types of Eigen's Matrix class (`https://eigen.tuxfamily.org/dox/group__TutorialMatrixClass.html`), and populated by the output of the local assembly kernels. Once all local data structures are filled, the dense linear algebra operations are performed and populated into the output tensor. The result is then passed onto Firedrake's global assembler (PyOP2).

We have discussed the possibility of providing a code listing in the paper, which would show some generated C++. We came to the conclusion that it would be unhelpful at best and, at worst, distracting away from the central message of the paper, which is the Slate abstraction itself.

4) "It is not totally clear how the output from TSFC is fed into the linear algebra compiler. Do you call the TSFC kernel, get the complete cell tensor, split it and then perform the

dense linear algebra operations? Or is everything 'interleaved' into one single cell tensor kernel by the linear algebra compiler? Or do you call multiple TSFC kernels, one for each sub-block and then perform the dense linear algebra operations?"

TSFC splits an assembly kernel for a mixed operator into separate kernels for each sub-block of the local tensor. Then the single kernel responsible for performing the dense linear algebra will first need to make multiple function calls to gather each local contribution. We shall make this more clear in Section 2 of the paper.

5) "The high-level problem setup in sections 1 and 2 is pretty terse. I don't think someone who has some knowledge of FEM but is not a real subject expert could get through this section. Some of the wording is quite heavy on jargon too, e.g. global data structure = sparse matrix? I appreciate you don't have time to do a deep-dive into FE, but a bit more text and some pointers to more detailed explanations (other Firedrake papers?) would be useful."

This is a very fair point. We will improve the sections to make the content more accessible to a general reader. In particular, we will steer away from too much jargon (or give concrete definitions) and provide appropriate references for the more technical details.

6) "Small notational comments: * You do not mention the bold symbol = vector function convention.

* Your convention of non-bold capitals being (local? - not sure) discrete linear operators (matrices and vectors) is also not mentioned, although K (the finite element cell) would break this 'convention'. I'd also note that K is used twice for different objects (the finite element cell and a matrix in eq.76) It would probably aid readability if it was possible to distinguish between matrices and vectors, global and local, especially when the operations become more complex in the later sections. There also seem to be (global? - not sure) discrete linear operators in bold."

We shall clarify our notation throughout the paper to better distinguish between local

and global tensors and their ranks (matrix vs vector). We will also avoid repeated use of particular characters, as you have pointed out ($K$ the cell vs $K$ the element tensor).

**References**

Kirby, Robert C., and Lawrence Mitchell. "Solver composition across the PDE/linear algebra barrier." SIAM Journal on Scientific Computing 40.1 (2018): C76-C98.

Rathgeber, Florian, et al. "PyOP2: A high-level framework for performance-portable simulations on unstructured meshes." 2012 SC Companion: High Performance Computing, Networking Storage and Analysis. IEEE (2012).

Homolya, Miklós, et al. "TSFC: a structure-preserving form compiler." SIAM Journal on Scientific Computing 40.3 (2018): C401-C428.

Please also note the supplement to this comment:
https://www.geosci-model-dev-discuss.net/gmd-2019-86/gmd-2019-86-AC2-supplement.zip

---

## Author Comment (AC3) · 28 Oct 2019

The authors would like to sincerely thank the referee for carefully reviewing the manuscript and providing helpful comments to improve the quality of the discussion.

1) "2.1. There are a few inaccuracies and omissions in the historical references for hybridization of mixed finite element methods, particularly in the introduction.

   2.1.1. The bibliographic reference to Brezzi and Fortin's book (from 1991, not 2012) is incorrect. The correct reference is:

   F. Brezzi and M. Fortin, Mixed and hybrid finite element methods, vol. 15

of Springer Series in Computational Mathematics, Springer-Verlag, New York, 1991.

The authors may also wish to cite the following, which is essentially an updated version of the Brezzi and Fortin book:

D. Boffi, F. Brezzi, and M. Fortin, Mixed finite element methods and applications, vol. 44 of Springer Series in Computational Mathematics, Springer, Heidelberg, 2013."

Thank you for pointing this out. We will correct the references in the introduction and include the updated book by Boffi, Brezzi, and Fortin (2013).

2) "2.1.2. Hybridization and static condensation of mixed methods was apparently first introduced in the following reference:

B. M. Fraejis de Veubeke, Displacement and equilibrium models in the finite element method, in Stress Analysis, O. Zienkiewicz and G. Holister, eds., Wiley, New York, 1965. Reprinted in Internat. J. Numer. Methods Engrg., 52 (2001), pp. 287–342."

The referee is absolutely correct. We will fix this omission in our discussion.

3) "2.1.3. Local post-processing appeared in Arnold and Brezzi (1985) for the hybridized RT method and in Brezzi, Douglas, and Marini (1985) for the hybridized BDM method. The authors frequently cite the former but not the latter when mentioning hybridization and post-processing of mixed methods. Also, the 1991 paper of Stenberg on post-processing is cited elsewhere in the paper but not in the introduction. I believe that these three papers should be cited in the relevant part of the introduction (p. 2, l. 18)."

This will be fixed in the revised paper.

4) "2.2. Section 2.1 is somewhat confusing and could be improved.

   2.2.1. The notation $a(c; v)$ originally made me think that $a(\cdot; \cdot)$ was a bilinear form. It took a few times through before I made sense of the notation and realized how linear, bilinear, etc., forms are specified. It might be helpful to include one or two concrete examples before introducing a "general form." This is done nicely in Section 2.1 of Alnaes et al. (2014), which I suggest that the authors emulate."

This is a very good point. We will rework this part of the paper and clarify our notation on bilinear forms to avoid confusion.

5) "2.2.2. The notation for the $\mathcal{I}^c$ and $\mathcal{I}^f$ integrals is also confusing. Presumably $c$ stands for "cell" and $f$ for "facet," but at first I thought $c$ was the coefficient function $c$ and $f$ was some source function, as in equation (10). Since the authors have been using $\mathcal{T}_h$ for cells and $\mathcal{E}_h$ for facets, perhaps a clearer notation would be to call these $\mathcal{I}^{\mathcal{T}}$, $\mathcal{I}^{\mathcal{E}, \circ}$, and $\mathcal{I}^{\mathcal{E}, \partial}$. A short sentence mentioning that the three $\mathcal{I}$s correspond to the contributions from the cells, internal facets, and boundary facets, respectively, would make this easier for the reader to understand."

Another good point and an excellent suggestion for improving our notation. We will incorporate these changes and provide further explanation of the individual terms.

---

## Author Response (AR1)

**Responses to reviewer comments on the manuscript: "Slate: extending Firedrake's domain-specific abstraction to hybridized solvers for geoscience and beyond."**

Thomas H. Gibson†

†*Email:* `t.gibson15@imperial.ac.uk`

November 20, 2019

**1 Anonymous Referee # 1**

The authors would like to sincerely thank the referee for the careful review and constructive comments for improving the manuscript.

*1) How is the .inv() command translated to generated code? Are you using LU/Cholesky or directly inverting the matrix? It is clear the former for the A.solve() command as the decomposition is passed, but not for .inv().*

This is a good question. As mentioned on page 6, Slate expressions will get transformed into C++ code using Eigen as the main linear algebra interface. By default, `A.inv()` gets translated to the corresponding Eigen call: `.inverse()`, which uses LU with partial pivoting for general matrices. This is also the default behavior of `A.solve(B)` when a factorization strategy is not provided. A complete list of factorization strategies which can be used with Slate are provided here: https://eigen.tuxfamily.org/dox/group__TopicLinearAlgebraDecompositions.html. This has been clarified in the revised manuscript (after Section 2.1.2).

*2) How would you deal in practice with non-linear problems solved using, e.g. a Newton-Krylov method. Is it possible to compose SLEPc and the Preconditioned Krylov Solvers? How would one setup SLEPc to call the code to recover the internal/local variables between Newton iterations?*

For global non-linear problems, one can use the technology provided in Section 4 of the paper together with PETSc or SLEPc non-linear solvers. Standard Newton, for example, requires the solution of the linearized Jacobian system for the linear updates. Each Newton iteration therefore requires a linear solve. In Firedrake, the Jacobian is constructed by differentiating the non-linear PDE (expressed in UFL), which generates yet another UFL expression for the Jacobian. The result is then used as the operator for the linear solver (KSP). Hybridization or static condensation can then be used by specifying the correct solver options. In practice, a single Newton iteration would look very similar to the Picard method described in Section 5.2. By design, all preconditioners presented in Section 4 are entirely composable with the PETSc library and can be applied in the usual way, even when nested inside of an outer non-linear method (e.g. `snes_type newtonls`, `ksp_type preonly`, `pc_type python`, `pc_python_type firedrake.SCPC`). This follows from the framework developed in [1].

*3) What does the generated Eigen code look like?*

Slate's linear algebra compiler generates templated C++ code conforming to PyOP2's application program interface, described in [2]. A typical generated kernel will take an output tensor, any coefficients, and possibly external data as arguments. The body of the kernel will make appropriate function calls to local assembly kernels, generated by Firedrake's form compiler TSFC [3]. Local data structures (element matrices/vectors) are derived types of Eigen's Matrix class (https://eigen.tuxfamily.org/dox/group__TutorialMatrixClass.html), and populated by the output of the local assembly kernels. Once all local data structures are populated, the dense linear algebra operations are performed and populated into the output tensor. The result is then passed onto Firedrake's global assembler (PyOP2).

We have discussed the possibility of providing a code listing in the paper, which would show some generated C++. We came to the conclusion that it would be unhelpful at best and, at worst, distracting away from the central message of the paper, which is the Slate abstraction itself.

*4) It is not totally clear how the output from TSFC is fed into the linear algebra compiler. Do you call the TSFC kernel, get the complete cell tensor, split it and then perform the dense linear algebra operations? Or is everything interleaved into one single cell tensor kernel by the linear algebra compiler? Or do you call multiple TSFC kernels, one for each sub-block and then perform the dense linear algebra operations?*

TSFC splits an assembly kernel for a mixed operator into separate kernels for each sub-block of the local tensor. Then the single kernel responsible for performing the dense linear algebra will first need to make multiple function calls to gather each local contribution. This made more clear in Section 2.

*5) The high-level problem setup in sections 1 and 2 is pretty terse. I dont think someone who has some knowledge of FEM but is not a real subject expert could get through this section. Some of the wording is quite heavy on jargon too, e.g global data structure  sparse matrix? I appreciate you dont have time to do a deep-dive into FE, but a bit more text and some pointers to more detailed explanations (other Firedrake papers?) would be useful.*

This is a very fair point. We have updated both Sections 1 and 2 to include some further explanation on finite element assembly and what the global data structures are.

*Small notational comments: * You do not mention the bold symbol = vector function convention. * Your convention of non-bold capitals being (local? - not sure) discrete linear operators (matrices and vectors) is also not mentioned, although K (the finite element cell) would break this convention. Id also note that K is used twice for different objects (the finite element cell and a matrix in eq.76) It would probably aid readability if it was possible to distinguish between matrices and vectors, global and local, especially when the operations become more complex in the later sections. There also seem to be (global? - not sure) discrete linear operators in bold.*

Notation throughout the entire paper has been updated and made more consistent.

**2    Anonymous Referee # 2**

The authors would like to sincerely thank the referee for carefully reviewing the manuscript and providing helpful comments to improve the quality of the discussion.

*2.1. There are a few inaccuracies and omissions in the historical references for hybridization of mixed finite element methods, particularly in the introduction.*
*2.1.1. The bibliographic reference to Brezzi and Fortins book (from 1991, not 2012) is incorrect. The correct reference is:*
*F. Brezzi and M. Fortin, Mixed and hybrid finite element methods, vol. 15 of Springer Series in Computational Mathematics, Springer-Verlag, New York, 1991.*
*The authors may also wish to cite the following, which is essentially an updated version of the Brezzi and Fortin book:*
*D. Boffi, F. Brezzi, and M. Fortin, Mixed finite element methods and applications, vol. 44 of Springer Series in Computational Mathematics, Springer, Heidelberg, 2013.*

Thank you for pointing this out. We have corrected the reference.

*2.1.2. Hybridization and static condensation of mixed methods was apparently first introduced in the following reference:*
*B. M. Fraejis de Veubeke, Displacement and equilibrium models in the finite element method, in Stress Analysis, O. Zienkiewicz and G. Holister, eds., Wiley, New York, 1965. Reprinted in Internat. J. Numer. Methods Engrg., 52 (2001), pp. 287342.*

The referee is absolutely correct. We have added an appropriate citation in the introduction.

*2.1.3. Local post-processing appeared in Arnold and Brezzi (1985) for the hybridized RT method and in Brezzi, Douglas, and Marini (1985) for the hybridized BDM method. The authors frequently cite the former but not the latter when mentioning hybridization and post-processing of mixed methods. Also, the 1991 paper of Stenberg on post-processing is cited elsewhere in the paper but not in the introduction. I believe that these three papers should be cited in the relevant part of the introduction (p. 2, l. 18).*

This has been fixed in the introduction.

*2.2. Section 2.1 is somewhat confusing and could be improved.*

*2.2.1. The notation $a(\boldsymbol{c}; \boldsymbol{v})$ originally made me think that $a(\cdot; \cdot)$ was a bilinear form. It took a few times through before I made sense of the notation and realized how linear, bilinear, etc., forms are specified. It might be helpful to include one or two concrete examples before introducing a "general form." This is done nicely in Section 2.1 of Alnaes et al. (2014), which I suggest that the authors emulate.*

This is a very good point. We have updated Section 2 to include more detailed constructions and definitions, following the convention of Alnaes et al. (2014).

*2.2.2. The notation for the $\mathcal{I}^c$ and $\mathcal{I}^f$ integrals is also confusing. Presumably c stands for "cell" and f for "facet," but at first I thought c was the coefficient function $\boldsymbol{c}$ and f was some source function, as in equation (10). Since the authors have been using $\mathcal{T}_h$ for cells and $\mathcal{E}_h$ for facets, perhaps a clearer notation would be to call these $\mathcal{I}^{\mathcal{T}}$, $\mathcal{I}^{\mathcal{E},\circ}$, and $\mathcal{I}^{\mathcal{E},\partial}$. A short sentence mentioning that the three $\mathcal{I}s$ correspond to the contributions from the cells, internal facets, and boundary facets, respectively, would make this easier for the reader to understand.*

Another good point and an excellent suggestion for improving our notation. We have incorporated this style in Section 2.

**3   Revised Manuscript**

All changes suggested by the reviewers have been incorporated. We have also made some clarifying edits to the discussion throughout Section 5. The numerical results themselves have not changed. A complete diff-summary of the paper is provided at the end of this document.

**References**

[revised manuscript text omitted]

$$\underline{A^K}\,\boldsymbol{A}^K_{\sim} = \begin{bmatrix} \boldsymbol{A}^K_{00} & \boldsymbol{A}^K_{01} & \cdots & \boldsymbol{A}^K_{0m} \\ \boldsymbol{A}^K_{10} & \boldsymbol{A}^K_{11} & \cdots & \boldsymbol{A}^K_{1m} \\ \vdots & \vdots & \ddots & \vdots \\ \boldsymbol{A}^K_{n0} & \boldsymbol{A}^K_{n1} & \cdots & \boldsymbol{A}^K_{nm} \end{bmatrix}. \tag{18}$$

The associated submatrix of (18) with indices $\boldsymbol{i} = (\boldsymbol{p}, \boldsymbol{q})$, $\boldsymbol{p} = \{p_1, \cdots, p_r\}$, $\boldsymbol{q} = \{q_1, \cdots, q_c\}$, is

$$\underline{A^K}\,\boldsymbol{A}^K_{\sim\,\boldsymbol{pq}} = \begin{bmatrix} \boldsymbol{A}^K_{p_1 q_1} & \cdots & \boldsymbol{A}^K_{p_1 q_c} \\ \vdots & \ddots & \vdots \\ \boldsymbol{A}^K_{p_r q_1} & \cdots & \boldsymbol{A}^K_{p_r q_c} \end{bmatrix} = \underline{A^K}\,\boldsymbol{A}^K_{\sim}\texttt{.blocks[}\boldsymbol{p},\boldsymbol{q}\texttt{]}, \tag{19}$$

[revised manuscript text omitted]

The mixed  formulation of (23)–(26) is arrived at by multiplying (23)–(24) by test functions and integrating by parts. The resulting finite element problem reads as follows: find  $(\boldsymbol{u}_h, p_h) \in U_h \times V_h$ satisfying

$$\left(\boldsymbol{w},\mu\boldsymbol{u}_h\right)_{\mathcal{T}_h} - \left(\nabla\cdot\boldsymbol{w},p_h\right)_{\mathcal{T}_h} = -\left\langle\boldsymbol{w}\cdot\boldsymbol{n},p_0\right\rangle_{\partial\Omega_D}, \quad \forall\boldsymbol{w}\in U_{h,0}, \tag{29}$$

$$\left(\phi,\nabla\cdot\boldsymbol{u}_h\right)_{\mathcal{T}_h} + \left(\phi,cp_h\right)_{\mathcal{T}_h} = \left(\phi,f\right)_{\mathcal{T}_h}, \quad \forall\phi\in V_h, \tag{30}$$

where $U_{h,0}$ is the subspace of $U_h$ with functions whose normal components vanish on $\partial\Omega_N$. The discrete system is obtained by first expanding the solutions in terms of the finite element bases:

$$\boldsymbol{u}_h = \sum_{i=1}^{N_{\boldsymbol{u}}} U_i\boldsymbol{\Psi}_i, \quad p_h = \sum_{i=1}^{N_p} P_i\xi_i, \tag{31}$$

where $\{\boldsymbol{\Psi}_i\}_{i=1}^{N_{\boldsymbol{u}}}$ and $\{\xi_i\}_{i=1}^{N_p}$ are bases for $U_h$ and $V_h$ respectively. Here, $U_i$ and $P_i$ are the coefficients to be determined. As per standard Galerkin-based finite element methods, taking $\boldsymbol{w} = \boldsymbol{\Psi}_j$, $j\in\{1,\cdots,N_{\boldsymbol{u}}\}$ and $\phi = \xi_j$, $j\in\{1,\cdots,N_p\}$ in (29)–(30) produces the discrete saddle point system:

$$\begin{bmatrix} \boldsymbol{A} & -\boldsymbol{B}^T \\ \boldsymbol{B} & \boldsymbol{D} \end{bmatrix} \begin{Bmatrix} \boldsymbol{U} \\ \boldsymbol{P} \end{Bmatrix} = \begin{Bmatrix} \boldsymbol{F}_0 \\ \boldsymbol{F}_1 \end{Bmatrix}. \tag{32}$$

where $\boldsymbol{U} = \{U_i\}_{i=1}^{N_{\boldsymbol{u}}}, \boldsymbol{P} = \{P_i\}_{i=1}^{N_p}$ are the coefficient vectors, and

$$\boldsymbol{A}_{ij} = \left(\boldsymbol{\Psi}_i,\mu\boldsymbol{\Psi}_j\right)_{\mathcal{T}_h}, \tag{33}$$

$$\boldsymbol{B}_{ij} = \left(\xi_i,\nabla\cdot\boldsymbol{\Psi}_j\right)_{\mathcal{T}_h}, \tag{34}$$

$$\boldsymbol{D}_{ij} = \left(\xi_i,c\xi_j\right)_{\mathcal{T}_h}, \tag{35}$$

[revised manuscript text omitted]

$$\left(\boldsymbol{A}_{11}^K - \boldsymbol{A}_{10}^K \left(\boldsymbol{A}_{00}^K\right)^{-1} \boldsymbol{A}_{01}^K\right) \boldsymbol{P}^K = \boldsymbol{F}_1^K - \boldsymbol{A}_{10}^K \left(\boldsymbol{A}_{00}^K\right)^{-1} \boldsymbol{F}_0^K - \left(\boldsymbol{A}_{12}^K - \boldsymbol{A}_{10}^K \left(\boldsymbol{A}_{00}^K\right)^{-1} \boldsymbol{A}_{02}^K\right) \boldsymbol{\Lambda}^K,\tag{48}$$

   followed by solving for $\left(\boldsymbol{U}^d\right)^K$:

$$\boldsymbol{A}_{00}^K \left(\boldsymbol{U}^d\right)^K = \boldsymbol{F}_0^K - \boldsymbol{A}_{01}^K \boldsymbol{P}^K - \boldsymbol{A}_{02}^K \boldsymbol{\Lambda}^K.\tag{49}$$

   Similarly, one could rearrange the order in which each variable is reconstructed.

4. If desired, the solutions can be improved further through local post-processing. We highlight two such procedures for $\boldsymbol{U}^d$ and $\boldsymbol{P}$, respectively, in Section 3.3.

 Figure 2 displays the corresponding Slate code for assembling the trace system, solving (45), and recovering the eliminated unknowns. For a complete reference on how to formulate the hybridized mixed system (41)–(43) in UFL, we refer the reader to Alnæs et al. (2014). Complete Firedrake code using Slate to solve a hybridizable mixed system is also publicly available in Zenodo/Tabula-Rasa (2019, "Code verification"). We remark that, in the case of this hybridizable system, (44) contains zero-valued blocks which can simplify the resulting expressions in (46)–(47) and (48)–(49). This is not true in general and therefore the expanded form using all sub-blocks of (44) is presented for completeness.

**3.2 Hybridization of discontinuous Galerkin methods**

The hybridized discontinuous Galerkin (HDG) method is a natural extension of discontinuous Galerkin (DG) discretizations. Here, we consider a specific HDG discretization, namely the LDG-H method (Cockburn et al., 2010b). Other forms of HDG that involve local lifting operators can also be implemented in this software framework by the introduction of additional local (i.e., discontinuous) variables in the definition of the local solver.

 Deriving the LDG-H formulation follows exactly from standard DG methods. All prognostic variables are sought in the discontinuous spaces $U_h \times V_h \subset [L^2(\Omega)]^n \times L^2(\Omega)$. Within a cell $K$, integration by parts yields:

$$(\boldsymbol{w}, \mu \boldsymbol{u}_h)_K - (\nabla \cdot \boldsymbol{w}, p_h)_K + \langle \boldsymbol{w} \cdot \boldsymbol{n}, \widehat{p} \rangle_{\partial K} = 0, \quad \forall \boldsymbol{w} \in U(K), \tag{50}$$

$$-(\nabla \phi, \boldsymbol{u}_h)_K + \langle \phi, \widehat{\boldsymbol{u}} \cdot \boldsymbol{n} \rangle_{\partial K} + (\phi, c p_h)_K = (\phi, f)_K, \quad \forall \phi \in V(K), \tag{51}$$

where $U(K)$ and $V(K)$ are vector and scalar polynomial spaces respectively. Now, we define the  numerical fluxes $\widehat{p}$ and $\widehat{\boldsymbol{u}}$ to be functions of the trial unknowns and a new independent unknown in the trace space $M_h$:

$$\widehat{\boldsymbol{u}}(\boldsymbol{u}_h, p_h, \lambda_h; \tau) = \boldsymbol{u}_h + \tau (p_h - \widehat{p}) \boldsymbol{n}, \tag{52}$$

$$\widehat{p}(\lambda_h) = \lambda_h, \tag{53}$$

where $\lambda_h \in M_h$ is a function approximating  $p$ on $\partial \mathcal{T}_h$ and $\tau$ is a positive stabilization function that may vary on each facet $e \in \partial \mathcal{T}_h$. We further require that $\lambda_h$ satisfies the Dirichlet condition for $p$ on $\partial \Omega_D$ in an $L^2$-projection sense. The full LDG-H formulation reads as follows. Find  $(\boldsymbol{u}_h, p_h, \lambda_h) \in U_h \times V_h \times M_h$ such that

$$\left(\boldsymbol{w}, \mu \boldsymbol{u}_h\right)_{\mathcal{T}_h} - \left(\nabla \cdot \boldsymbol{w}, p_h\right)_{\mathcal{T}_h} + \left\langle [[\boldsymbol{w}]], \lambda_h \right\rangle_{\partial \mathcal{T}_h} = 0, \quad \forall \boldsymbol{w} \in U_h, \tag{54}$$

$$-\left(\nabla \phi, \boldsymbol{u}_h\right)_{\mathcal{T}_h} + \left\langle \phi, \left[\left[\boldsymbol{u}_h + \tau \left(p_h - \lambda_h\right) \boldsymbol{n}\right]\right] \right\rangle_{\partial \mathcal{T}_h} + \left(\phi, c p_h\right)_{\mathcal{T}_h} = \left(\phi, f\right)_{\mathcal{T}_h}, \quad \forall \phi \in V_h, \tag{55}$$

$$\phantom{=}_{\partial \mathcal{T}_h \backslash \partial \Omega_D} \left\langle \gamma, \left[\left[\boldsymbol{u}_h + \phantom{}_{\partial \Omega_D} \tau \left(p_h - \lambda_h\right) \boldsymbol{n}\
[revised manuscript text omitted]
 \left[\!\left[ \underset{\sim}{\boldsymbol{n}}^\perp \boldsymbol{w} \cdot \underset{\sim}{\boldsymbol{u}_h}^\perp \right]\!\right], \tilde{\underset{\sim}{\boldsymbol{u}}}_h^\perp \right\rangle_{\partial\mathcal{T}_h}$$

$$- \left(\nabla \cdot \boldsymbol{w}, g\left(D_h + b\right) + \frac{1}{2}|\boldsymbol{u}_h|^2\right)_{\mathcal{T}_h} = 0, \quad \forall \boldsymbol{w} \in \underset{\sim}{U}_h, \tag{A1}$$

$$\left(\phi, \frac{\partial D_h}{\partial t}\right)_{\mathcal{T}_h} - \left(\underset{\sim}{\nabla \phi}, \underset{\sim}{\boldsymbol{u}_h} D_h\right)_{\mathcal{T}_h} + \left\langle \left[\!\left[ \underset{\sim}{\phi \boldsymbol{u}_h} \right]\!\right], \tilde{\underset{\sim}{D}}_h \right\rangle_{\partial\mathcal{T}_h}, = 0, \quad \forall \phi \in V_h, \tag{A2}$$

where $\tilde{\cdot}$ indicates that the value of the function should be taken from the upwind side of each facet. The discretisation of the velocity advection operator is an extension of the energy-conserving scheme of Natale and Cotter (2017) to the shallow-water equations.

The time-stepping scheme follows a Picard iteration semi-implicit approach, where predictive values of the relevant fields are determined via an explicit step of the advection equations, and corrective updates are generated by solving an implicit linear system (linearized about a state of rest) for $(\Delta \boldsymbol{u}_h, \Delta D_h) \in \boldsymbol{U}_h \times V_h$, given by

$$(\boldsymbol{w}, \Delta \boldsymbol{u}_h)_{\mathcal{T}_h} + \frac{\Delta t}{2}\left(\boldsymbol{w}, f\Delta \boldsymbol{u}_h^\perp\right)_{\mathcal{T}_h} - \frac{\Delta t}{2}\left(\nabla \cdot \boldsymbol{w}, g\Delta D_h\right)_{\mathcal{T}_h} = -R_{\boldsymbol{u}}[\boldsymbol{u}_h^{n+1}, D_h^{n+1}; \boldsymbol{w}], \quad \forall \boldsymbol{w} \in \underset{\sim}{U}_h, \tag{A3}$$

$$\left(\underset{\sim}{\phi, \Delta D_h}\right)_{\mathcal{T}_h} + \frac{H\Delta t}{2}\left(\underset{\sim}{\phi, \nabla \cdot \Delta \boldsymbol{u}_h}\right)_{\mathcal{T}_h} = -R_D[\boldsymbol{u}_h^{n+1}, D_h^{n+1}; \phi], \quad \forall \phi \in V_h, \tag{A4}$$

where $H$ is the mean layer depth, and $R_{\boldsymbol{u}}$ and $R_D$ are residual linear forms that vanish when $\boldsymbol{u}_h^{n+1}$ and $D_h^{n+1}$ are solutions to the implicit midpoint rule time discretization of (A1)–(A2). The residuals are evaluated using the predictive values of $\boldsymbol{u}_h^{n+1}$ and $D_h^{n+1}$.

The implicit midpoint rule time discretization of the  nonlinear rotating shallow water equations (A1)–(A2) is:

$$\left(\boldsymbol{w}, \boldsymbol{u}_h^{n+1} - \boldsymbol{u}_h^n\right)_{\mathcal{T}_h} - \Delta t\left(\nabla^\perp\left(\boldsymbol{w} \cdot \boldsymbol{u}_h^{*\perp}\right), \boldsymbol{u}_h^{*\perp}\right)_{\mathcal{T}_h} + \Delta t\left(\boldsymbol{w}, f\boldsymbol{u}_h^{*\perp}\right)_{\mathcal{T}_h}$$

$$+ \Delta t\left\langle \left[\!\left[ \underset{\sim}{\boldsymbol{n}}^\perp \boldsymbol{w} \cdot \underset{\sim}{\boldsymbol{u}_h}^{*\perp} \right]\!\right], \tilde{\underset{\sim}{\
[revised manuscript text omitted]